# Intensified paraglacial slope failures due to accelerating downwasting of a temperate glacier in Mt. Gongga, southeastern Tibet Plateau

Yan Zhong[1,2], Qiao Liu[1], Matthew Westoby[3], Yong Nie[1], Francesca Pellicciotti[4], Bo Zhang[5], Jialun Cai[5], Guoxiang Liu[5], Haijun Liao[1,2], Xuyang Lu[1]

[1] Institute of Mountain Hazards and Environment, Chinese Academy of Sciences, Chengdu 610041, China
[2] College of Resources and Environment, University of Chinese Academy of Sciences, Beijing 100049, China
[3] Department of Geography and Environmental Sciences, Engineering and Environment, Northumbria University, Newcastle upon Tyne NE1 8ST, UK
[4] Swiss Federal Institute for Forest, Snow and Landscape Research WSL, 8903 Birmensdorf, Switzerland
[5] Department of Surveying and Geo-Informatics, Faculty of Geosciences and Environmental Engineering, Southwest Jiaotong University, Chengdu 611756, China

*Correspondence to*: Qiao Liu (liuqiao@imde.ac.cn)

**Abstract.** Topographic development via paraglacial slope failure (PSF) represents a complex interplay between geological structure, climate, and glacial denudation. Southeastern Tibet has experienced amongst the highest rates of ice mass loss in High Mountain Asia in recent decades, but few studies have focused on the implications of this mass loss on the stability of paraglacial slopes. We used repeat satellite- and UAV-derived imagery between 1990 and 2020 as the basis for mapping PSFs from slopes adjacent to Hailuogou Glacier (HLG), a 5 km-long monsoon temperate valley glacier in the Mt. Gongga region. We observed recent lowering of the glacier tongue surface at rates of up to 0.88 m a$^{-1}$ in the period 2000 to 2016 whilst overall paraglacial bare ground area (PBGA) on glacier-adjacent slopes increased from 0.31±0.27 km$^2$ in 1990 to 1.38±0.06 km$^2$ in 2020. Decadal PBGA expansion rates were ~0.01 km$^2$ a$^{-1}$, 0.02 km$^2$ a$^{-1}$, and 0.08 km$^2$ in the periods 1990-2000, 2000-2011, and 2011-2020 respectively, indicating an increasing rate of expansion of PBGA. Three types of PSF, including rockfalls, sediment-mantled slope slides, and headward gully erosion, were mapped, with a total area of 0.75±0.03 km$^2$ in 2020. South-facing valley slopes (true left of the glacier) exhibited more destabilization (56% of the total PSFs area) than north-facing (true right) valley slopes (44% of the total PSFs area). Deformation of sediment-mantled moraine slopes (mean 1.65-2.63±0.04 cm d$^{-1}$) and an increase in erosion activity in ice-marginal tributary valleys caused by a drop in local base level (gully headward erosion rates are 0.76-3.39 cm d$^{-1}$) have occurred in tandem with recent glacier downwasting. We also observe deformation of glacier ice, possibly driven by destabilisation of lateral moraine, as has been reported in other deglaciating mountain glacier catchments. The formation, evolution, and future trajectory of PSFs at HLG (as well as other monsoon-dominated deglaciating mountain areas) are related to glacial history, including recent rapid downwasting leading to the exposure of steep, unstable bedrock and moraine slopes, and climatic conditions that promote slope instability, such as very high seasonal precipitation and seasonal temperature fluctuations that are conducive to freeze-thaw and ice segregation processes.

# 1 Introduction

Climatic warming induced permafrost degradation and glacier shrinkage have altered the thermal, hydraulic, and mechanical properties of their adjacent terrains, where the destabilizing slopes increase the risk of hazards in high mountain regions (Krautblatter and Leith, 2015). The thinning and retreat of mountain glaciers expose new, unstable landscapes which are susceptible to rapid geomorphological change (i.e., with high entropy). Sparsely vegetated or unvegetated drift-mantled slopes are particularly susceptible to modification by gravitational, aeolian and fluvial processes and can be rapidly reworked by debris flows, rock avalanches and slope-wash (Ballantyne, 2003; Ballantyne, 2002; Deline et al., 2015a; Eichel et al., 2013). Glacier downwasting can destabilize slopes through undercutting, and progressive reductions in ice loading (debuttressing) of ice-marginal slopes can result in rock stress release and, in turn, instability. The observed permafrost degradation is also exacerbating these local changes in topography (Gruber et al., 2017). The four predominant modes of slope response in deglaciating catchments are: (1) large-scale catastrophic rock slides and rock avalanches (Kirkbride and Deline, 2018; Fischer et al., 2010); (2) ice-contact slope movements (McColl and Davies, 2013); (3) periodic small-scale rock topples or rockfalls (Cook et al., 2013); and (4) deep-seated gravitational creep (Deline et al., 2015b; Ballantyne et al., 2014). These responses provide a useful framework for examining the processes and geomorphological consequences of slope adjustment during or following deglaciation. These responses can be further divided into two types: *rock slope failures* and *sediment slope failures*, which can be described collectively as *paraglacial slope failures* (PSFs). As one major cold region geomorphic process following glacial and permafrost-related slope instability, PSFs encompass failure of steep rock walls and lateral moraine slopes following glacier downwasting (Church and Ryder, 1972; Fickert and Grüninger, 2018) and are widely distributed in deglaciating or deglaciated landscapes.

PSFs constitutes a key mechanism for rapid degradation of recently glacial landscapes, as they contribute to the disaggregation of large portions of valley sides and the transformation from U-shaped to V-shaped valleys. PSFs can transport a considerable volume of debris onto glacier surfaces (Smith et al., 2020), thereby facilitating the accumulation of the supraglacial moraine. More recently, a study in the Himalaya measured the sediment from lateral moraines increasing in debris thickness (sediment slope failures) by 0.08 m a$^{-1}$ of the glacier surface (van Woerkom et al., 2019). The rocks and debris from a catastrophic rock avalanche (rock slope failures) in southeast Tibet covered the entire glacier surface and the wider area around and downstream (Kääb et al., 2021). They (lateral moraine and bordering rock walls) encompass two of the three potential sources of supraglacial debris (the other is basal erosion, (Reheis, 1975; Boulton, 1978)). Because thin debris covers enhance ablation, whilst thicker covers suppress it (Fyffe et al., 2020), debris supply dynamics ultimately affect glacial ablation and meltwater production (Rowan et al., 2018). In addition, debris deposited by PSFs can be (re) mobilised into mass flows and other hazard cascades that directly threaten human life, property, and infrastructure, and can affect the freshwater quality (Hewitt et al., 2011; Reznichenko et al., 2012). A high-profile example is the 2021 Chamoli mass flow, northern India, that was the rock-ice

avalanches transformed into a catastrophic debris flow, causing the destruction of two hydropower projects and more than 200 casualties (Shugar et al., 2021).

Monsoonal temperate glaciers on the southeastern Tibetan Plateau (SETP) have experienced higher rates of mass loss and ice thinning compared to other glacierized regions in High Mountain Asia in past decades (Neckel et al., 2017; Brun et al., 2017). In addition to being seismically active, the combination of high rates of glacier ablation and abundant monsoonal precipitation and steep topography make the SETP highly susceptible to cryospheric hazards (Yao et al., 2019a), such as glacial debris flow, ice/snow avalanche, glacier-related landslide, and glacial lake outburst flood (Fan et al., 2019; Cheng et al., 2010; Xu et al., 2012). For example, at least four glacier-related landslides (1990-2018, (Liu et al., 2010; He et al., 2008; Pan et al., 2012; Cao et al., 2019; Liu et al., 2018; Liu and Liu, 2010)), and 12 glacial lake outburst floods (1931-2014, (Liu et al., 2019b; Yao et al., 2014) have been recorded across the SETP, and it has been suggested that more than half of the debris flows recorded in the SETP were cryogenic (Hu et al., 2011). Mass movements resulting from large-scale slope failures routinely form temporary valley-blocking natural dams (Liu et al., 2019c), behind which large lakes can develop and eventually outburst. Glacier destabilization itself was also one of the concerned sources of cascade hazard events, such as historically reported surge induced catastrophes (Zhang, 1985) and recent ice avalanche induced debris flow that blocks the Yarlung Tsangpo river (Chen et al., 2020; Kääb et al., 2021). These cascading geo-hazards have caused severe economic losses or some heavy casualties to SETP local communities.

With populations and economies expanding, it is anticipated that the magnitude and frequency of cryospheric hazards in the SETP will increase in near future (Yao et al., 2019b). Since the accelerating glacial mass loss and permafrost degradation due to climate warming will continue to destabilize the paraglacial landscapes (McColl, 2012). A growing body of work focuses on the destabilization of paraglacial hillslopes in European Alps (Curry et al., 2006; Kirkbride and Deline, 2018), Southern Alps (Cody et al., 2020; McColl, 2012), Cordillera Blanca (Andes) (Emmer et al., 2020) and Central Himalayas (van Woerkom et al., 2019), which have reported more on slope monitoring. Although they have recorded the glacier retreat and thinning and listed it as the main factor of slope instability, case studies combining the monitoring of multi-dynamic of glaciers and paraglacial slope and reporting for the transient condition of paraglacial slopes in the deglaciating monsoonal temperate glaciers in SETP are clearly lacking. We hypothesize that glacier dynamic change, including retreat, downwasting, and slowing, can explain the (prone to) instability of paraglacial slope. Meanwhile, in the monsoon-dominated temperate glacier regions, a climate of warm-wet synchronization is the primary precondition driving the paraglacial rock and sediment slope failure.

To this end, we select the Hailuogou Glacier (HLG, a monsoon temperate glacier in Mt. Gongga that has been well documented and observed since the early 20th century) as a detailed case site to study the interactions between glacier shrinkage and paraglacial (rock and sediment) slope adjustments during the past decades. We combine visual analysis of historical satellite imagery and imagery acquired from an unpiloted aerial vehicle (UAV), with in-situ geomorphological observations and

measurements to: (i) map and quantify the spatiotemporal variability of PSFs around the lower tongue of HLG based on time-series image comparison; (ii) classify the PSFs, analyse their topographic conditions, and discuss their possible physical mechanisms in monsoon-dominated temperate glacier regions; (iii) discuss the geomorphic and environmental effects of these slope processes by exploring the possible linkage between climate change, glacier downwasting and PSFs as accelerated glacial and paraglacial denudation.

## 2 Study area

HLG (29°58.4′N, 101°91.6′E) is one of the largest debris-covered valley glaciers in the Mt. Gongga region (Fig. 1). It is a ~13 km-long, east-facing monsoon temperate glacier, has an area of 24.7 km², and extends in altitude from 2990 m to 7556 m a.s.l. The glacier is ~300-500 m in width, and the wider valley is deeply incised below the accumulation area, where the glacier broadens to ~6 km in width. The upper section of the ice tongue (below the icefall) has a surface gradient of ~10° and is connected to very steep (~55°) lateral slopes on both sides. At a distance of ~1.5 km from the base of the icefall (middle section of the ice tongue) the glacier surface steepens to ~12° and becomes highly crevassed. At a distance of 2 km, the glacier turns its flow direction toward the northeast and maintains a surface gradient of ~13° for the remaining 3 km (lower section of the ice tongue) stretching to the end of the glacier tongue. Paralleling with the lower section of the ice tongue, the height of lateral moraines to the current glacier surface is about 165 m, which is comparatively higher than the further upper glacier (~86 m at the middle section of ice tongue).

The terminus of HLG has retreated more than 2 km since the LIA (Su and Shi, 2002) and this retreat has accelerated from 12.7 m a$^{-1}$ between 1966–1989 to 27.4 m a$^{-1}$ between 1998–2008. Between 1966 and 2009, the mean ice surface elevation of its lower tongue had lowered at a rate of 1.1±0.4 m a$^{-1}$ (Zhang et al., 2010). Due to suitable local thermal (warm) and moisture (wet) climatic conditions, vegetation is able to quickly establish itself on the exposed proglacial bare ground in glacier forelands and lateral moraine slopes, and even on some supraglacial debris-covered areas that are largely stagnant. The LIA preglacial zone (2980-2800 m) is characterized by extremely fast primary vegetation succession and hosts an integrated community ranging from cold-adapted herbaceous species to *Abies fabri* (conifer pine) forest. *Abies fabri* forest patches are also seen along both lateral sides of the ice tongue on the accumulated lateral moraine, which has been relatively stable since the LIA.

Due to its low altitude, easy accessibility, and adjacency to the forests and hot springs, HLG and its surrounding areas have been exploited for tourism since 1987. A glacier ropeway connects the two lateral moraines and transports visitors across the lower part of the ice tongue and provides access to a viewpoint on the southern side of the valley (S2 on Fig. 1) from where the ice tongue can be overlooked and accessed via trail. Between 2000 and 2011, the number of annual visitors increased from 44,000 to 384,000 (Zhu, 2015), and in 2019 the number reached ~2,852,600 (Haiguanju, 2020). The increasing popularity of the glacier comes at a time when tourism facilities and infrastructure are becoming frequently disturbed due to paraglacial

landslides, debris flows, and flash floods (Xu et al., 2007; Cai et al., 2021). Beyond the objectives outlined above, there is a need to more fully understand the nature of slope failures in the vicinity so that these failures might be better anticipated, and their potential impacts mitigated where possible.

## 3 Data and methods

### 3.1 Paraglacial slope failures: mapping and classification

Vegetation colonisation of newly exposed bare ground is very fast (1-2 years) in the HLG catchment. Like many monsoonal temperate glaciers, the lower part of the HLG tongue and its adjacent slopes are located below the local tree line, and so areas of new bare ground caused by slope destabilisation are relatively easy to detect; we call these areas paraglacial bare ground area (PBGA). Due to the strong influence of the monsoon, the HLG is often covered with thick clouds in summer, rendering 75% of the historical summer satellite image archive unusable. However, we successfully screened and synthesized several

Landsat or Sentinel-2 cloudless images in 1990, 2000, 2016, and 2018 using the Google Earth Engine platform. A total of 8 satellite images acquired between 1990 and 2020 (Table 1) including Sentinel-2 (10m), RapidEye (5m), PlanetScope (3m), were used to extract the PSFs. We extracted non-vegetated areas by classifying the land surface as vegetated or unvegetated using the Normalized Difference Vegetation Index (NDVI), which was derived using reflected red and near-infrared band values of satellite images. Shadows caused by the relative angle between the sun, ground objects, and sensors will reduce the

value of NDVI, but by adjusting the threshold, the vegetated and non-vegetated areas can still be differentiated. Therefore, we used cloudless images in multiple seasons (spring, autumn, and summer) to automatically extract PSFs boundaries. Of the seasonal data, the PSF boundaries extracted from images acquired in the northern hemisphere summer generally suffered less from the terrain shadow than other seasons and can be used directly after validation, and the boundaries extracted from other seasons need to be manually corrected. PBGAs were then extracted by excluding the glacier-covered areas (GCAs). For better

inter-annual comparisons, we then manually removed those patches and areas where the PBGA had changed due to vegetation colonisation in some paraglacial slopes and in all glacier foreland to keep the remaining unstable slope areas as final mapped PSFs. According to the principles of systematization, standardization, operability and scalability, the PSF are classified combining the knowledge of the basic slope material composition, erosion type, visual shape, and event magnitude obtained from field investigation and UAV data in HLG, and the classifications proposed by Ballantyne (2002), McColl (2012), Jarman

(2006), and Hungr et al. (2014). A classification standard for PSF systems is proposed, which is divided into two major categories: rock slope failure and sediment slope failure; and five sub-categories: rock avalanches, rockfall, deep-seated gravitational slope deformations, sediment-mantled slope slide, and gulley headward erosion (Tab. S1).

To enable more detailed mapping of recent PSF extents, UAV sorties were flown on 31 August 2016, 07 June 2017, and 15

May 2019 using a DJI Phanton 4 Pro UAV, and 19 August 2018 using a DJI Phanton 4 RTK UAV. Aerial photographs were co-aligned using Structure from Motion (SfM) software ContextCapture Center Master (version 4) to create a time series of

orthomosaic aerial images at 0.1m resolution and with a mean RMS error of 0.01 m + 1 ppm[*] (RMS) in XY. However, some of our UAV surveys included limited coverage of off-glacier terrain (i.e., Section 3.2), and so we could not map PSFs on all ice-marginal slopes (Fig. S1). To enable co-registration of UAV images with satellite imagery, we selected 54 features (e.g., exposed bedrock, trees, or building vertices) from stable ground areas (Fig. S1) for use as ground control points (GCP) and which were identifiable on both PlanetScope and UAV images. The altitude (z) of each GCP was then extracted from the ALOS PALSAR DEM. The final mean (and maximum) RMS error of the GCPs was 1.22 (1.98) pixels, equating to a mean xy horizontal error of 0.15 m. An additional error of ±0.5 pixels was estimated to include all the uncertainties of visual interpretation and automatically extracted areas (i.e. multiply linear error and perimeter, e.g. the linear error of Landsat images are 15m, Sentinel 2 images 5 m and PlanetScope images ~1.56 m) (Salerno et al., 2012; Haritashya et al., 2018). Our workflow is summarized in Fig. 2.

### 3.2 Slope movement and headscarp erosion rates

To quantify the rate of slope movement between 2016 and 2019, we selected 20-40 points (trackable features such as rocks, paths, or infrastructures) on a series of UAV-mapped slopes (B1-B4 in Fig. S2, Fig. S3). We manually tracked the location of each point in QGIS software and calculated their mean diurnal horizontal displacement. For areas exhibiting no discernible slope movement, but where we observed headscarp erosion of slope instabilities, we quantified this rate of headscarp erosion. Based on PSF boundaries delineated on five higher resolution satellite images from 2002 to 2019 (Google Earth SPOT-5 imagery for 2002, RapidEye images 2011, 2014, and 2015, and PlanetScope image 2019), we calculated the mean distance and annual retreat rates for these headscarps using the QGIS Average Nearest Neighbour Analysis procedure.

### 3.3 Glacier dynamics

Because the evolution of paraglacial slopes is closely related to glacier dynamics (Ballantyne and Benn, 1994; Ballantyne, 2002), we analysed the geomorphological processes and implications of retreat, thinning and slowdown of the HLG and their relationships with PSF development. Detailed mapping of the retreat and thinning of the HLG over past decades has been well documented and reported (Li et al., 2010; He et al., 2008; Liu et al., 2018; Zhang et al., 2010; Liu et al., 2010). We extend the boundary of glacier terminus retreat to 2020 using some newly archived satellite images (Fig. 1 and Table 1). We employed the TopoDEM (1966) reported by Zhang et al. (2010), the Shuttle Radar Topography Mission (SRTM, 2000) 30m DEM (Farr et al., 2007), and a calculated 2016 DEM based on the 2000-2016 surface elevation change rate from Brun et al. (2017) to analyse the surface elevation changes of HLG between 1966 and 2016. Surface elevation changes along five profile-lines in longitudinal (A-A' along the glacier central flow line) and transverse (B-B', C-C', D-D' and E-E' perpendicular to the glacier central flow line) on the lower part of HLG (Fig. 1) were examined and compared. The four transverse lines intersected approximately every type of PSF (Section 4.2) and a sufficient distance is maintained in each transverse profile for comparison.

---

[*] 1 ppm means the error has a 1mm increase for every 1 km of movement from the drone.

For ice flow dynamics, the earliest observations of surface velocity of the HLG were during 1982-1991 field expeditions, during which the location changes of several stakes along the ice tongue were repeatedly measured. Unfortunately, the original coordinate data of those measurements (e.g., the precise geographic locations of the stakes) was not offered and results are published only as a velocity isoline map with point velocity value indicated. Using 38 SAR images acquired by PALSAR-1/2 satellites from 2007 to 2018, Liu et al. (2019a) extracted annual surface displacement velocities of the HLG between 2007 and 2018. We compared the long-term ice flow velocity changes of the HLG (also checked along the five profile-lines mentioned above), based on three-periods results of 1982-1983, 2007-2011 and 2014-2018. The velocity in 1982-1983 is based on a published velocity map (Fig. S4), which was derived via extrapolating of in-situ measurements of stake locations acquired using total station. Glacier surface velocities for the period 2007-2011 and 2014-2018 were derived from ALOS/PALSAR satellites using feature tracking method offered by Liu et al. (2019a).

### 3.4 Meteorological data

Meteorological conditions nearby the HLG were diagnosed to discuss their possible forcing mechanisms and/or impacts on the development of PSFs. Daily air temperature and precipitation (1988/01/01--2018/12/31) were observed by the Gongga Alpine Ecosystem Observation and Experiment Station (Gongga Mt. Station, 3000 m a.s.l., 2 km to the glacier terminus) with a standard automatic weather station (AWS) installed and managed by the Chinese Ecological Research Network (CERN; *http://www.cern.ac.cn/*). We also collected the manually observed daily temperature data (observed at 2:00 am, 8:00 am, 2:00 pm, 8:00 pm every day by a staff of Mt. Gongga Station) to compensate for the missed AWS data between July and September 2017 due to equipment inspection and between 29 September and 15 October 2018 due to electrical power failures. During the period 1988-2018, the observed mean annual temperature at 3000 m a.s.l. was 4.5 ℃. Rain gauge recorded an average annual number of 314 days and a mean annual amount of 1912 mm precipitation, of which 88% amount was concentrated in ablation seasons (April to October) and ~42% was concentrated in summer (July to September).

### 4 Results

### 4.1 Retreating, thinning and deceleration of the Hailuogou Glacier

Glacier outlines in 2002, 2016 and 2020 (Fig. 1c) were manually delineated based on high resolution images (Google Earth SPOT-5 imagery for 2002, UAV images for 2016, and PL imagery for 2020). Comparison of these outlines shows that the glacier tongue area has reduced from $2.30\pm0.06$ km$^2$ to $1.95\pm0.02$ km$^2$ (average $0.09\pm0.03$ km$^2$ a$^{-1}$) between 2016 and 2020 compared with the period 2002-2016, when it reduced from $2.65\pm0.01$ km$^2$ to $2.30\pm0.06$ km$^2$ (average $0.03\pm0.01$ km$^2$ a$^{-1}$). Between 2002 and 2016, the position of the glacier terminus showed a moderate retreat ($6\pm0.44$ m a$^{-1}$); whereas during the following four years (2016-2020), the terminus retreated more than $220\pm1.56$ m (~$54\pm0.39$ m a$^{-1}$). Field observations indicate

that the current glacier terminus area was no longer covered by thicker debris than the previous status, and frequent collapse along the subglacial outlet channel became the major backwasting process causing the accelerated terminus retreat.

The lower part of the HLG tongue also showed continuous narrowing as it retreats due to ice thinning. Ice surface elevation and velocity along five profile lines (Fig. 3) show that the ice tongue has experienced substantial ice loss and slowdown over the past decades. The longitudinal line (A-A') along the glacial flow direction runs through the entire ablation zone. Between 1966 and 2000, we observed negligible to small changes in the surface elevation of the upper ice tongue (-0.5~0 m a$^{-1}$, between 3400 to 3700 m a.s.l), with a mean lowering the rate of -0.11 m a$^{-1}$. However, the ice tongue below 3200 m a.s.l (3~4.5 km to

the base of the icefall) thinned remarkably as the highest lowering rate up to -2.2 m a$^{-1}$ near the glacier terminus. In contrast, the thinning rate of the upper ice tongue has greatly increased between 2000 and 2016, with the surface elevation change rate remaining at around -2.0 m a$^{-1}$. On the other hand, the lower part of the ice tongue (below 3400m) shows a decreasing thinning rate towards the terminus. For transverse profiles B-B' and C-C', rates of surface elevation lowering were much higher during 2000-2016 (-1.8 m a$^{-1}$) than 1966-2000 (-0.3~0 m a$^{-1}$). In contrast, profiles D-D' and E-E' showed less surface lowering over

the past 50 years, with thinning rates remaining around -0.6 m a$^{-1}$ and -0.9 m a$^{-1}$, respectively.

Analysis of ice surface velocities during three periods (1982-1983, 2007-2011, and 2014-2018) in the past 38 years (Fig. 3c) indicates that the lower ablation area of HLG has experienced a gradual slowdown, with the rate of this deceleration decreasing of glacier (Fig. 3c, profile A-A') and with increasing proximity to the lateral margins (Fig. 3c, profiles B-E). Ice surface

velocities along all transverse cross sections have gradually decreased, with the greatest deceleration occurring along with profiles B-B' to D-D' (mean rate of 0.26 m a$^{-1}$) and less along the E-E' section (mean rate of 0.06 m a$^{-1}$). Note that remarkable decelerations that happened across the whole transverse profile of D-D' during our observation periods have led the ice tongue below almost stagnating (with a velocity less than 0.1 m a$^{-1}$) during 2014-2018.

## 4.2 Paraglacial slope failures

Mapped extents of the PBGAs between 1990 and 2020 were presented in Fig. 4. The PBGA show continual expansion during the observation period, increasing from 0.31±0.27 km$^2$ in 1990 to 1.38±0.06 km$^2$ in 2020, with a total areal increase of 1.07±0.32 km$^2$. Its decadal expansion rates increased from ~0.01 km$^2$ a$^{-1}$ in 1990-2000 and 0.02 km$^2$ a$^{-1}$ in 2000-2011, to ~0.08 km$^2$ a$^{-1}$ in 2011-2020, which was approximately proportionate to the annual decrease in the GCA. Due to a transient rockfall below the icefall that occurred between September and November 2018, the PBGA increased significantly, with an expansion

rate up to ~0.11 km$^2$ a$^{-1}$ (10.43%) from 2018-2019 within one year. In general, the north-facing slope (located on the true right side of the glacier) exhibits more stability than south-facing slope, where 56% of the total PBGA was mapped.

According to the classification method in Tab. S1, we identified three typical styles of paraglacial slope failure within the PBGA, with a total area of 0.75±0.03 km$^2$ in 2020 (Fig. 5a):

A.  Rockfall, 0.06±0.01 km$^2$ in 2020, 8% of the total PSF area.

B.  Sediment-mantled slope slide, 0.37±0.01 km$^2$ in 2020, 49% of the total PSF area.

C.  Gully headward erosion, 0.32±0.01 km$^2$ in 2020, 43% of the total PSF area.

Based on calculated surface slope map from the 2016 DEM (Fig. S5) as well as field verifications (Fig. S6), the Type B and C PSFs (with a mean surface slope of 29° and 32°, respectively) usually show lower slopes in their unstable areas than stable areas (40-60°). The mean slope of the Type A PSF is about 54°, which is within the range of the stable slope. Below we will present a detailed descriptive analysis for each type of PSF observed at HLG and provide a summary of the dimension and typology for each specific PSF identified in Fig. 5a in Table 2.

**4.2.1 Rockfalls (paraglacial slope failure type A)**

Different magnitude rockfalls are common at exposed steep bedrock terrain on both sides of HLG. A recent large rockfall event (Type A, Fig. 5b) was occurred around October 15 2018 originating from a south-facing paraglacial slope high-hanging above the upper HLG ice tongue (4000 m a.s.l., ~450 m high from the glacier surface). The rockfall deposit and the existing glacier debris cover are slightly different in colour and grain size and can thus be clearly delineated on the glacier surface, both

on field photograph and on remote sensing imagery (Fig. 5b and Fig. S7). The deposition area of the rockfall covers a projected area of ~47,000 m$^2$, stretching a vertical height of 380 m and slope length of 472 m (runout length ~365 m). Based on the extent of the main detachment zone, the deposition thickness (3-5 m estimated during the field visit) and area (Fig. 5b), total release volume of the rockfall is 1,000-10,000 m$^3$. We also observed several suspected major rockfall scars on the lower part of the valley wall in close proximity to this scar deposit (i.e., up to 260~400 m above the glacier surface; Fig. 5b and 5c)

suggesting that numerous similar scale rockfalls have occurred in this locality. Since the rock surface has been partially weathered and most of the bedrock has been covered by vegetation such as mosses and shrubs, we speculate that these rockfalls may have occurred earlier and have not been documented by remote sensing images or literature.

Except for the 2018 large rockfall, we identified at least 15 fresh small-magnitude rockfalls in view of their evident talus

depositions based on UAV images and field investigation (Fig. 5a, Fig. S7). These small-scale rockfalls are found located on both sides of the lower paraglacial slopes and have a source area ranging from 59 to 3028 m$^2$. Their disintegrating rocks were located between 3100 and 3500 m a.s.l. with a mean vertical distance of about 110 m from their lowest breakpoints to the glacier surface. Unfortunately, due to their small sizes and without continuous observations (e.g., time-lapse photography), we cannot determine the exact time when they happened. These collapsed rocks falling on the glacier surface with different

magnitude obviously are one of the important inputs for supraglacial debris on the lower part of the HLG ice tongue.

#### 4.2.2 Sediment-mantled slope slide (paraglacial slope failure type B)

We identified evidence of four major sediment-mantled slope slides (Type B, B1-B4, Fig. 5a). Combined, these features cover an area of ~370,000 m² (2020-PL) and are located on both north- and south-facing valley sides. The 2D area of Type B PSFs steadily expanded through four processes: glacier downwasting exposing more of the slope, lateral expansion of the failure mass, and headward expansion from retrogressive failure or degradation of the scarp, over our monitoring period (Fig. 4). The total area increased by 297,000 m² and the area expansion rate was 10.16% $a^{-1}$ from 1990 to 2020.

B1 (3500 m a.s.l, Fig. S2a) is located on a north-facing lateral moraine slope at a distance of 1.5 km from the base of the icefall and is 114 m wide, 1035 m long, and 112,424 $m^2$ in area. Between 2017 and 2019 the exposed area of this feature B1 slope increased by 12,125 $m^2$, or a percentage increase of 12.09 %. A landslide with an area of 53,000 $m^2$ is included as a nested feature within B1 (48.28% of B1 area), and evidence of superimposed gullying is also found on the eastern, western, and northern segments of this feature. The UAV images in 2016 did not cover the B1 area completely, so we extracted 23 common tie points across UAV images from 2017 to 2019 to manually quantify the surface displacement rate; the slope showed downslope displacement of 1.96±0.04 cm $d^{-1}$, and a maximum of 2.08±0.04 cm $d^{-1}$ in the period 2017-2018.

Feature B2 (3150 m a.s.l, Fig. S2b) is the largest type B PSF and is located on the true left, south-facing moraine slope close to the glacier terminus (3240-3120 m a.s.l.). It is fan-shaped with a width of ~262 m, a length of 644 m, a height of ~120 m, and a total area of 122,738 $m^2$. In contrast to B1, this feature is characterized as a single landslide without any nested features (e.g., gullies). Visual analysis of satellite imagery shows that detectable slope movement began around the year 2000, which is consistent with the onset of accelerated lowering of the glacier surface in this region and accompanying debuttressing of ice-marginal topography. We selected 33 tie points evenly distributed in the B2 landslide area on the four-year UAV images (2016-2019) to monitor its surface displacement, which revealed that the landslide has moved by an average of 2.63±0.04 m $a^{-1}$ in the UAV monitoring period, and the affected area has increased by 7,414$m^2$ via headscarp erosion. The highest rate of displacement was 4.32 cm $d^{-1}$ and occurred between 2017-2018. Between 2011 and 2019, the collapsed area of B2 has expanded by 35,000 m², with an expansion rate of 4.99% $a^{-1}$. The landslide has a vertical glide distance of 48 m, and a horizontal displacement of 70 m. In the period 2017-2018 landslide activity forced the closure of a zigzag trail path (Fig. 5e) which was used for accessing the glacier.

Across the valley from B2, we identify landslide B3 (3100 m a.s.l, Fig. 5f, Fig. S2c), which is 805 m in length, 132 in width, and 63,241 $m^2$ in area and, like B2, is located on a lateral moraine close to the glacier terminus. The landslide has been effectively divided into five zones, delineated by four gullies (Fig. 5f). Analysis of satellite imagery shows that the landslide began to develop around 2013, when a headscarp is first detectable. Feature tracking (33 tie points) analysis on repeat UAV

images shows that the landslide has moved down with an average rate of 1.65±0.04 cm d$^{-1}$ between 2016 and 2019, with the highest rate of 1.97±0.04 cm d$^{-1}$ occurring in the period 2017-2018.


Feature B4 (Fig. S2d) is also located on the north-facing (true right) lateral moraine slope of the glacier, approximately 3.5 km from the base of the icefall. It is ~103 m in height, 993 m in length, and covers an area of 73,270 m$^2$. Geomorphologically, this feature is the most complex of the type B PSFs, in that it represents a transition slope that it exhibits landslides in two distinct zones at either end (covering an area of 18,244 m², or 24.90% of the sediment-mantled moraine slope in this area) and also

exhibits gully headward erosion (i.e., PSF Type C). Feature tracking analysis of 37 tie points indicates that the landslide moved downslope at an average rate of 1.66±0.04 cm d$^{-1}$ between 2016 and 2019, with a maximum rate of 2.63±0.04 cm d$^{-1}$ in 2017-2018.

### 4.2.3 Gully headward erosion (paraglacial slope failure type C)

The three major paraglacial gully headward erosion areas (C1-C3, Fig. 5a) are located at the intersections between three

tributary streams which are fed by seasonal snowmelt and the main trunk of the HLG ice tongue. They have a total area of ~324,000 m² in 2019. The UAV images did not cover the entire areas of Type C slopes, therefore we used five PL, RapidEye, and Google Earth images from 2002, 2011-2019 to supplement these data and monitor the development of these features (Fig. S8). Our mapping results show that the total area of these three gullies has expanded by ~93,000 m² between 2011 and 2019, with an annual expansion rate of 5.02% a$^{-1}$.


C1 is located on the northern slope of HLG at an altitude of 3200 m (Fig. 5g; Fig. S8a). In 2019 it had a length of 849 m, a width of 312 m and, an area of 139,135 m². It is adjacent to a major tributary stream that drains the northern side of the valley, and which intersects (and bisects) the true left lateral moraine where the glacier turns to flow eastward to its terminus. Field inspection of exposed surfaces at C1 reveals that it comprises a block of thick debris and sand deposits with some finer material,

which was previously colonized by vegetation before being denuded by flowing water. In the period 2011-2015, the area of the gully expanded upslope at a rate of 10.44 m a$^{-1}$, and by 2019 the location of headward erosion was more clearly concentrated in the meltwater tributary channel (Fig. S8a). Between 2011 and 2019, the average upward denudation rate of the feature was 3.39 cm d$^{-1}$ (12.20 m a$^{-1}$) and the exposed area had increased by 45,449 m² or 48.59% from 2011 (Tab. 2).

Gully C2 is situated on the southern slope (true right lateral moraine) of HLG at about 3200 m a.s.l (Fig. 5h; Fig. S8b) and in 2019 extended ~160 m in width, ~270 m in length, and 24,248 m² in area. The exposed area of the gully expanded upslope along the path of a meltwater-fed tributary stream, with a gradient of ~36$^o$ close to the HLG moraine, and which originates from Hailuogou No. 3 Glacier (29° 32.41' N, 101 58.05' E). This tributary stream connects directly to the subglacial water system of HLG. The gully is most actively eroding upstream of the point at which it connects to HLG, with an average upward

denudation rate of 0.76 cm d$^{-1}$ (2.77 m a$^{-1}$) between 2002 and 2019, producing an increase in the exposed area of 11,923 m² in the same period.

Gully C3 is also located on the southern slope of HLG at 3500 m a.s.l (Fig. 5i; Fig. S8c), around ~800 m west of gully C2. The gully is formed by a tributary stream which is fed by meltwater from Hailuogou No. 2 Glacier (29° 32.82' N, 101° 56.92'
E), which became detached from HLG sometime after the 1930s (Liu and Zhang, 2017) and has since retreated to the edge of ice scarped ridge. The gully is filled with glaciofluvial sediment and is deeply incised. Like C2, meltwater also flows into HLG glacier subglacially. Since 2002, expansion of the sediment-mantled area of this gully has occurred on the right (east) side of the gully and has recently begun to expand up along the left (west) bank of the channel. We expand the area of the gully to incorporate adjacent unvegetated lateral moraine, which also shows evidence of gullying and headward erosion. The total area
of C3 increased by 14.3%, from 109,500 m² in 2002 to 160,474 m² in 2019, with an average upward denudation rate of 1.15 cm d$^{-1}$ (4.20 m a$^{-1}$) between 2002 and 2019.

# 5 Discussion

## 5.1 Possible forcing mechanism of paraglacial slope failures at Hailuogou glacier

Many potential factors, including glacial history (notably downwasting and debuttressing), rock structure (e.g. joint
distribution), seismicity, and short- and long-term local meteorology have been generally considered as preconditioning, preparatory or triggering factors that combine to produce a PSF (McColl, 2012). In subsections below we refer to these factors to discuss their relative importance in forcing the different types of PSF that we observed at HLG.

### 5.1.1 Rockfall

Small-scale rockfalls are commonly sourced from exposed bedrock walls on both sides of the HLG ice tongue. These exposed
bedrock walls are generally granite rocks characterised by well-developed vertical joints (Fig. S6 and S7), which make the slope more prone to failure or more likely failure under given triggers (Draebing and Krautblatter, 2019). In addition to slope oversteepening as a result of glacial erosion (Herman et al., 2021), the 'fresh' exposure of these bedrock slopes as a result of glacier thinning can also accelerate weathering processes and result in a reduction in rock strength (Matsuoka, 2008). Intensification of freeze-thaw conditions during deglaciation may have led to more enhanced rockfall activity since the winter
season at HLG is cold enough to cause water to freeze within rock fractures and pore spaces. Conversely, the warmth and humidity of the summer monsoon season can also enhance rates of chemical weathering of exposed rock walls (Li et al., 2019), and a reduction in rock strength. However, we cannot determine the frequency and magnitude of these small-scale rockfalls in this study because the specific time of their occurrence was unknown, and we likely observe a degree of small-scale event censoring as smaller rockfall scars progressively enlarge via detachment of subsequent rockfalls. We therefore focus on the

discussion of possible forcing mechanisms of a larger scale rockfall event that occurred in 2018, which we could confirm its occurring time (around 15 Oct. 2018) and estimate its magnitude from UAV surveys.

The 2018 rockfall (Fig. 5b) represents a rapid non-ice contact rock collapse adjacent to an area of the glacier that has experienced remarkable (80-100 m) thinning since its Little Ice Age (LIA) maximum (Fig. 5c). More widely in the Mt. Gongga

region, glacial and paraglacial landform assemblages display a strong influence of Late Pleistocene and Holocene glaciations, the most striking example of which are the deeply incised trunk and tributary valleys formed by higher rates of glacier erosion (Liu et al., 2009), glaciofluvial activity, and monsoonal precipitation; as such, paraglacial slopes are strongly modified by local glacial history, making this an important preconditioning and preparatory factor for rockfall. Anecdotally, small-scale rock collapses occur all year round on both sides of HLG (Fig. 5a; Fig. S7), especially from those rock slopes exhibiting vertical

jointing. Our field investigations indicate that the main detachment zone of the 2018 rockfall has no faults but is highly jointed, which facilitates liquid water ingress and hydraulic erosion and is also conducive to promoting fracturing via freeze-thaw and ice segregation processes (Rodríguez-Rodríguez et al., 2018) which gradually loosens the rock block and might also be considered a trigger in causing blocks to eventually break away from the rock valley walls (Fischer et al., 2010).

The annual mean temperature was relatively low in 2018 (4.5°C) compared to the preceding four years (5.0°C, Fig. 6a). Fig. 6b shows the antecedent rainfall in the five to ten days before the rockfall occurred (5-20 Oct 2018) and compares antecedent rainfall statistics between five years (2014-2018). The mean daily precipitation between 5 and 20 Oct 2018 is 5.89 mm which is significantly higher than 2014, 2015, and 2017, although not significant compared with 2016, the value is still high. Additionally, the differences in the daily mean temperature between September to October 2018 and September to October

2014-2017 were relatively large (Fig. 6c; from September to October 2018, a large amount of daily temperature data was missing from the automatic observation and was therefore offset by manual observations (MO), as mentioned in Section 3.4 above); the daily mean temperature was between 0.4 to 5.3°C from 01 to 09 October 2018 compared to a range of 3.6-12.92°C for the preceding four years. From 20 to 29 September, the mean daily temperature was similar to the preceding four years with a range of 6.6-13.3°C. The temperature lapse rate (0.0065°C $m^{-1}$ in humid air, (Hemond and Fechner, 2015)) have shown

that when the temperature at the Gongga Mt. Station (3000 m a.s.l.) drops to about 6°C, the temperature on the mountain at 4000 m a.s.l. approaches 0°C and freezing of water begins. We, therefore, speculate that the abundant precipitation in the early days caused a large amount of liquid water to ingress the rock fissure, then the frost heave caused the rock mass to eventually rupture (e.g. (Hartmeyer et al., 2020)). We finally suggest that abnormal precipitation and low temperature in 2018 may have acted as a preparatory factor and triggering factor for the observed rockfall, respectively.


Rockfall and other mass movements can be triggered by seismic activity, which can also act as a preparatory factor through its ability to cause rock damage (Huggel et al., 2007). The HLG region is seismically active, and in the last five years has recorded 9 seismic events ranging from 2.9-3.8 in magnitude, all of which occurred in the period 2016-2017 (China Earthquake

Administration, *https://www.cea.gov.cn/*). Although no significant seismic events were observed immediately prior to the 2018
rockfall, rock mass damage associated with the historical seismic activity may have acted as a preparatory factor for the 2018
rockfall and other mass movements from deglaciated slopes in the wider catchment.

### 5.1.2 Sediment-mantled slopes slide

Type B PSFs are largely associated with lateral moraine instability and collapse. Elsewhere (north-facing slopes, Fig. S9) in
the catchment we observe glacially smoothed and polished bedrock surfaces with steep inclination angles (a product of the
geological, glacial, and climatic history of the catchment), and we infer that the failure of these unconsolidated slopes is caused
by glacier downwasting and debuttressing (Cody et al., 2020), combined with a material angle of repose which is lower than
the inclination of the underlying bedrock surface; sediment-mantled slopes showing evidence of instability typically have a
slope angle of ~29°, whilst the inclination of exposed bedrock slopes close to the elevation of the present-day glacier surface
is ~45°. Whilst historically the glacier has essentially 'propped up' the lateral moraines, rapid reduction in the elevation of the
glacier surface has directly contributed to slope instability, which is increasing in speed, corresponding to findings in van
Woerkom et al. (2019).

Debris flow may also be a preparatory factor causing slope instability. A gully developing process was observed within Type
B failures (as mentioned in Section 4.2.2 above). Based on 17 years (2002-2019) RS monitoring of the B3 slope (Fig. S10),
for example, we found that the surficial debris flow occurred before the hillslope movement. Firstly, some debris flows
occurred, forming 3 gullies, and no slide was evident in 2002. Secondly, the debris flow gullies gradually expanded, the number
increased to 5, the slope slid slightly, and the slide cracks were starting to form around in 2013. Finally, the debris flow gullies
increased to 6 and expanded further (the largest gully is about 130 m wide at its widest point), the slope slide significantly (up
to 2 cm d$^{-1}$ between 2017 and 2018; Fig. S2), and the cracks increased in response to rapid glacier downwasting in 2019, which
is different from the discovery of the Fox glacier in New Zealand by Cody et al., (2020).

The regional and local climate, which is characterized by abundant precipitation during the monsoon season, enables rapid
succession of vegetation on deglaciated slopes. However, although this vegetation is extensive in its coverage, particularly on
slopes immediately above the lateral moraines, it mostly comprises species with a shallow rooting depth that do not have a
strong capacity to increase slope stability, especially at larger scales; shrub vegetation growing on the unstable B2 and B3
slopes are relatively young and do not mitigate deep translational failure, whereby unconsolidated sediment slides along the
interface between the overlying moraine and steep underlying bedrock. We also observed that the displacement of all type B
PSFs reached their maximum rate during the period 2017-2018, and upper parts of the slope failures speed up to the same
extent as the lower parts (Fig. S3), which may in part be related to increased air temperature (promoting enhanced glacier
downwasting) and precipitation extremes in 2018.

### 5.1.3 Gully headward erosion

Glacier downwasting and meltwater from deglaciating tributary catchments enhance erosion of ice-marginal gullies and contribute to the expansion of actively eroding slopes in these landscapes; headward erosion of sidewall tributary gullies occurs as the local base level (i.e., the HLG glacier surface) falls (Williams and Koppes, 2020; Schiefer and Gilbert, 2007), and this effect is illustrated clearly at PSF C1 (Fig. 5h and Fig. S8a).

The upstream sediments were transported and accumulated along the gullies to both sides of the glacier, with a fast erosion rate, for instance, the part with the largest change in C1 area moved about 150 m in both horizontal and vertical directions. Therefore, we argue that surface fluvial erosion and glacier debuttressing simultaneously trigger the instability of the type C slopes (Dusik et al., 2019).

All the three major type C slopes are developed along the tributary streams, in which settings slope slides are usually limited but surface fluvial erosion plays the primary role. There should exist a seasonal plus or enhanced headward erosion rates, i.e., during the monsoon rainstorm seasons. However, we still could not track this process due to the lack of higher resolution observations in the current study. Nevertheless, it is suggested that this type of paraglacial adjustment is still in active process based on the currently observed situation at both lateral sides of the HLG. As the increase of the exposed area and the erosional baseline, a period of accelerated sediment/debris flux to the surface or base of the glacier is expected in the following years.

### 5.2 Paraglacial geomorphology process in deglaciating monsoon temperate environments

This study has demonstrated that paraglacial slopes in deglaciating monsoon temperate environments are characterised by a range of geomorphological process responses, leading to high rates of landscape denudation. The combination of rapid glacier downwasting exposing unstable moraine and oversteepened bedrock walls with well-developed vertical jointing, and a climate that is conducive to the presence of both freeze-thaw processes and enhanced rates of chemical weathering due to the warmth and high humidity of the summer monsoon season provide favourable conditions for the development of paraglacial slope failures. The following subsection revisits the overall hypothesis of the study, that the climate of warm-wet synchronization is the primary precondition which driving the paraglacial rock and sediment slope failure (Section 5.2.1), and the glacier dynamic change can explain the (prone to) instability of paraglacial slope (Section 5.2.2); and the other implications of paraglacial geomorphology process (Section 5.2.3).

### 5.2.1 Impact of the monsoonal climate conditions

Climatic conditions (e.g., fluctuation of temperature and precipitation) can directly affect slope stability (McColl, 2012; Coe, 2020). Meteorological observations at the Gongga Mountain Station (Fig.1) show that the regional mean annual temperature has increased by about 1 °C over the past 30 years (Fig. 6d). Diurnal (Fig. 6c) and annual (Fig. 6a) variation of temperature

may affect the slope freeze-thaw process (Fischer et al., 2012; Curry et al., 2006). Abundant rainfall (5.89 mm in Section 5.1.1) in the monsoon season leads to saturation of glacier-adjacent slopes, and whilst it is well-established that vegetation cover can provide a stabilising effect on regolith, saturation of vegetated slopes nevertheless results their failure. Small-scale debris flows in the monsoon season are common, which is also the main reason for the occurrence of nested processes in Type B (Fig. 5f, S8b, and S11). The combination of extreme temperature and precipitation ranges can also trigger slope instability (as mentioned in Section 5.1.1 above - the relatively low temperature and high precipitation in 2018 may be the cause of rockfall).

**5.2.2 Paraglacial slope process interaction with the dynamics of temperate glacier**

Despite a slowdown in recent decades (Fig. 3c), the flow velocity of HLG is still higher than most mountain glaciers in Himalaya, Tian Shan, and inner Tibet (Bhushan et al., 2018; Wang et al., 2016; Zhang et al., 2010; Ke et al., 2013). Previous work has quantified the evolution of the glacier's subglacial and englacial hydrology, and revealed a highly efficient hydrological system that is maintained in both summer and winter months (Liu and Liu, 2010); an abundance of meltwater and subglacial and englacial debris (which we infer because of the debris-covered nature of the glacier, e.g. (Miles et al., 2021)) act as effective "grinding tools" for glacial abrasion. The estimated erosion rate is 2.2-11.4 mm $a^{-1}$ (Based on the current glacier thickness and flow velocity, a conservative estimate of the glacier bedrock erosion depth is about 1-5 m) which is consistent with other temperate glaciers but higher than the continental glaciers (0.1-1.0 mm $a^{-1}$)(Liu et al., 2009). A high erosion rate contributes to valley incision and the steepening of valley flanks, which increase in exposed area and angle and therefore likely to become more prone to instability.

Warming-induced downwasting of HLG serves as a preparatory or triggering factor for PSFs. Monsoon-temperate glaciers, particularly those which are steep, are characterised by higher rates of mass turnover (Oerlemans, 1997), and well-developed en- and sub-glacial drainage systems (Fountain and Walder, 1998) which can lead to a faster dynamic response to changes in air temperature and precipitation compared to glaciers found in colder climates . Glacier downwasting can trigger the failure of Type B and C PSFs (as mentioned in Section 5.1.2 and 5.1.3 above). According to Fig. 3 and 4, in profile D-D', the D' side (true left of the glacier) has been thinned (with a mean rate of -1.0 m $a^{-1}$) and slowed (the mean daily velocity changed by -0.27 m $d^{-1}$) more significantly than that on the D side (true right of the glacier; with a mean rate of 0 m $a^{-1}$ and the mean daily velocity changed by -0.14 m $d^{-1}$, respectively) in the latest study period, resulting in the slope failure on D' side has been faster. Bare, exposed glacier-adjacent slopes may increase the absorption and emission of long-wave thermal radiation and contribute to locally enhanced ablation and surface lowering of glacier ice. Profile E-E' showed uniform thinning (Fig. 3b, -1.0 m $a^{-1}$), and we detected landslides on both sides of the glacier. For both profiles B-B' and C-C', the thinning rate changed from ~0 m $a^{-1}$ in 1966-2000 to ~-2 m $a^{-1}$ in 2000-2016 (Fig. 3b), which may be related to the inner stress adjustment in the glacier as the loss of ice downglacier results in upstream acceleration. Both sides of the profile B-B' are mainly steep bedrock, with a limited accumulation of lateral moraine materials, such that type A is the dominant PSF. The glacier downwasting is mainly used as

a preparatory factor for the slope instability here. The relationship between climatic change, glacier downwasting, and PSF is shown in Fig. 7.

According to Fig 4b, the PBGA was equivalent to ~11% of the GCA in 1990 and increased to ~69% in 2020, thereby representing an ample source of sediment that could either be delivered to the supraglacial environment, or, through remobilisation, bypass the glacier itself and be transferred to the proglacial environment or beyond. The total area of all observed rockfalls covered 62216 m$^2$ (volume class: $10^4\sim10^5$ m$^3$), which is 2.3% of supraglacial debris-cover area (which is 2.71 km$^2$, according to Scherler et al., (2018)) of HLG. Due to the high supraglacial debris-cover of HLG (11%), the contribution rate of rockfalls observed in this study is lower than that of European Alps and Southern Alps of New Zealand rockfalls of the same size class (Fischer et al., 2012). We also observed a limited number of materials that are delivered to the supraglacial environment through direct roll (Fig. S11 (a); the sediments are more dispersed compared to rockfalls), collapse (Fig. S11 (b)), and debris flow (Fig. S11 (c); faster changes). Among them, two collapsed parts of the B2 slope were quantified by using UAV imagery in the 718 days from 2016/8/31 to 2018/8/19, the collapsed area increased by 94% in total, with an increased rate of 8 m$^2$ d$^{-1}$. These sediment delivery process increase the range and thickness of the supraglacial debris-cover, which in turn affects the surface energy balance for melting and therefore downwasting rate of the glacier, and which will in turn influence meltwater production. Additionally, some processes (e.g., Type C slope failures) can also deliver materials directly to the sub-glacial environment via runoff (Fig. S11 (d)). Their contribution to glacial sediments is hard to quantify, but it is clear that this process does result in a long-term and steady sediment supply to the glacier.

Because ice has a lower density than rock (generally 1:3), glacier-adjacent slope instabilities can cause deformation of glacier ice; this geomorphological phenomenon has been identified in the Southern Alps of New Zealand (McColl and Davies, 2013), where mountain glaciers occupy valleys which bear similarities in steepness and width to our study catchment. Field evidence from the monitoring of the B2 slope by UAV images for four consecutive years (Fig S6b) indicates that the glacier in this vicinity has narrowed or squeezed. The direction of the glacier movement at the boundary with B2 changes as slope B2 advances, which causes the original arc of the glacier boundary to migrate. The greatest change in the location of the glacier boundary bordering slope B2 is during the period 2017-2018, which is consistent with the fastest displacement speed (~4.32 cm d$^{-1}$) of the slope during the wider monitoring campaign. Our results may imply that the weight of the moraine may be sufficient to deform the glacier ice, as hypothesized by McColl and Davies (2013).

### 5.2.3 Other geomorphological and environmental implications

Similar to other studies (Dunning et al., 2015; Glueer et al., 2020; Hedding et al., 2020), our analyses show a temporal and spatial component to PSF development following rapid deglaciation of the HLG. Following many thousands of annual thermo-hydromechanical loading cycles, which promote rock damage (Grämiger et al., 2017; Glueer et al., 2020) the bedrock slopes bordering HLG are well-prepared for failure, and this response is beginning to manifest as specific triggering events occur

and/or mechanical rock mass thresholds are crossed. Similarly, unconsolidated lateral moraines show signs of increasing deformation and translational movement in response to glacier downwasting and debuttressing; we hypothesize that this will continue until critical angles of repose are reached, which will be followed by vegetation colonization and soil development
(Eichel et al., 2018).

With respect to the terrain aspect, we found that south-facing slopes show more evidence of instability, both in terms of magnitude and frequency. Firstly, it could be related to the differences in the availability of material (Fig. S9; i.e. asymmetrical deposition of glacial drift and moraine construction on either side of the valley), with more sediment on the south-facing slopes
and slope angles (Fig. S9). Secondly, it could also be related to the differences in glacier surface altitude and ice thinning rate, which is particularly obvious compared to the C-C', D-D' and E-E' profiles. It can be seen from profile D-D' in Fig. 3 a, b that the altitude and thinning rate of the glacier under the south-facing slope are much greater than that on the opposite side (ice thinning rate of difference was 1.5 m a$^{-1}$), which may be caused by the lower surface temperature due to the topographic shading on the glacier areas under the north-facing slope (Liao et al., 2020). Therefore, PSFs and glaciers downwasting show
the same temporal and spatial movement. This observed pattern further indicates that there is a direct connection between glacier downwasting and PSFs occurrence. Additionally, the unstable slopes are more likely to occur in areas with lower slopes (all-around 30°, as mentioned in Section 4.2 above) is contrary to the general conclusion of the many landslides study - the steeper the slope, the stronger the shear stress and the lower the factor of safety (McColl, 2015), which may be related to the slope material and the failure process.


Additionally, the unstable slopes are more likely to occur in areas with lower slopes (all-around 30°, as mentioned in Section 4.2 above) is contrary to the general conclusion of the many landslides study - the steeper the slope, the stronger the shear stress and the lower the factor of safety (McColl, 2015), which may be related to the slope material and the failure process. We suggest a conjectural paraglacial slope evolution and slope sediment delivery model of HLG (Fig. 8) to explain this slope
anomaly and to explore the logical relationship between the three types of PSFs. After the glacier was downwasting from the initial state (Stage I), the upper of the steep moraine slope quickly destabilized, making the entire slope slow, corresponding to funding in van Woerkom et al. (2019), Ballantyne (2013), and Cully et al. (2006); then the moraine slope slowly slides down (Stage II; which were observed in Type B and part of Type C). During the exposure of the moraine, vegetation may be colonized, or gullies may be formed by debris (or water) flows washing away (stage II a); when the gullies gradually expand,
headward erosion may occur in the upper of the gullies (Stage II b; which were observed in Type C). Sediments at the base of the slope or fell onto the glacier surface are transferred again as the glacier moves (Stage III); until the slope is steepest (>50°) when moraine has been removed and bedrock is completely exposed to the ground and may collapse through some disturbances, finally (Stage IV; Type A).

# 6 Conclusion

We used repeat UAV and satellite remote sensing imagery as the basis for identifying and analysing the evolution of three styles of paraglacial slope failure at HLG between 1990 and 2020 and explore these results in the context of their potential driving mechanisms. Over this period the glacier terminus has retreated ~510 m (17 m a$^{-1}$), its surface has down-wasted by -0.88 m a$^{-1}$, with thinning observed over 83.50% of the ablation zone study area, and the glacier velocity has slowed from a mean of 0.32 m d$^{-1}$ (period 1982 to 1983) to 0.11 m d$^{-1}$ (2014 to 2018). Rapid downwasting of the glacier surface has exposed

oversteepened ice-marginal slope topography, which shows evidence of the overall paraglacial bare ground area (PBGA) increased from 0.31±0.27 km$^2$ in 1990 to 1.38±0.06 km$^2$ in 2020; and of widespread instability in the form of (A) rockfall from bedrock slopes, (B) slide of unconsolidated lateral moraines, and (C) increased erosion activity in tributary valleys, with a total area of 0.75±0.03 km$^2$ in 2020. South-facing valley slopes (true left of the glacier) exhibited more destabilization (56% of the total PSFs area) than north-facing (true right) valley slopes (44% of the total PSFs area). Set against a background of

frequent, small-scale rockfalls that are anecdotally recorded in the area, a large rockfall occurred in Autumn 2018. Non-ice-contact rockfalls in deglaciating catchments are a well-established, short- to long-term response of paraglacial slopes following glacier downwasting and rock slope exposure, however, analysis of antecedent meteorological conditions suggests that abundant precipitation and low temperature may have served to prepare, and perhaps trigger this rock slope failure. Deformation of sediment-mantled moraine slopes (mean 1.65-2.63±0.04 cm d$^{-1}$) and an increase in erosion activity in ice-

marginal tributary valleys caused by a drop in local base level (gully headward erosion rates are 0.76-3.39 cm d$^{-1}$) have occurred in tandem with recent glacier downwasting. The PSFs and even the whole paraglacial bare ground area provides an ample source of sediment for the glacial environment. We speculated that the occurrence of some unstable slopes in the lower slope angle areas may be related to the slope material and the failure process. These two factors combined with glacier dynamics, local topography and other factors, contribute to spatial heterogeneity of paraglacial geomorphology process. Considering our

discussion and initial hypothesis, the high erosion rate of HLG associated with the high flow velocity (0.11 m d-1 or 40 m a-1) contributes to valley incision and steepening of valley flanks which increase in exposed areas and therefore likely explain that the slope is more prone to instability after accelerating downwasting of the glacier. In general, the formation, evolution, and current status of these typical PSFs are generally related to the history of glacier dynamics and paraglacial geomorphological adjustments, and also influenced and/or disturbed by the fluctuation of air temperature/precipitation and

their combinations. Longer-term monitoring will provide a clearer picture of the feedbacks between (accelerating) glacier downwasting, climatic conditions, and paraglacial landscape response in this data-poor region.

**Data availability**

The paraglacial slope failure shape files can be requested by email from the author zhongyan19@mails.ucas.ac.cn

## Author contributions

QL and XL initiated the underlying research project and obtained the funding. QL developed the research goal and designed the study. QL, YN, BZ, JC, HL and GL performed the primary UAV data. QL and YZ conducted and analysed the data. YZ, QL, MW, YN and FP wrote the paper.

## Competing interests

The authors declare that they have no conflict of interest.

## Acknowledgements

This work was funded by the NSFC Project (41871069) and the Sichuan Science and Technology Programs (2021JDJQ0009, 2020JDJQ0002). The authors gratefully acknowledge the PlanetLab for provision of PlanetScope and RapidEye imagery, and U.S. Geological Survey for Landsat satellite images. MW and FP acknowledge funding from Royal Society-Newton Fund project 'Understanding glacier and hydrological changes in the Tibetan Plateau using high-resolution monitoring and modelling'. We thank the editor Michael Krautblatter and the two reviewers Samuel McColl and Jan Henrik Blöthe for their comments and suggestions, which helped to improve this article.

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

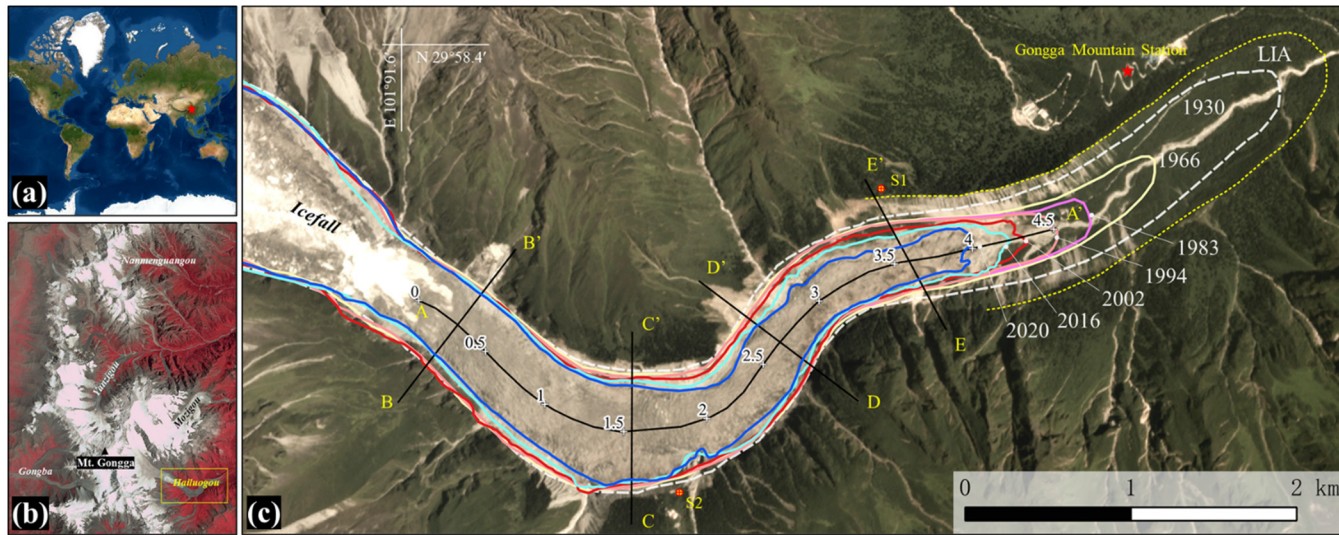


**Figure 1: Map showing the location of Mt. Gongga (red star), the background image is from ESRI's world basemap (a), several large glaciers around the peak of Mt. Gongga based on Landsat image (b), and retreating history of HLG since the Little Ice Age based on PlanetScope imagery (c). S1 and S2 are tourist sightseen stands on both side of lateral moraines, and the red star indicates the location of Gongga Mountain Station. Longitudinal (A-A') and transverse (B-B', etc.) lines are set to examine the variation of flow**
**velocities and surface elevations (Fig. 3) of the ice tongue.**

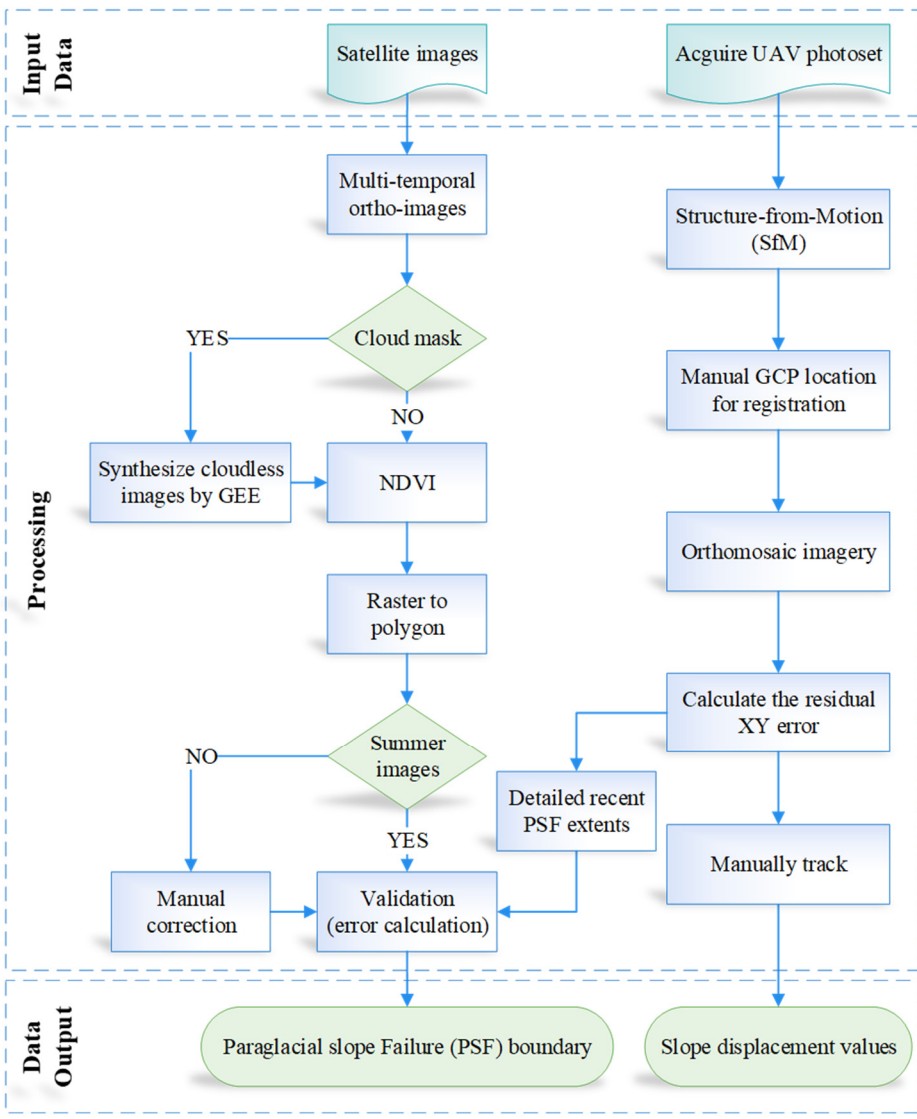

**Figure 2: Workflow for mapping the PSFs (changes) and tracking the displacement from multi-temporal image data.**

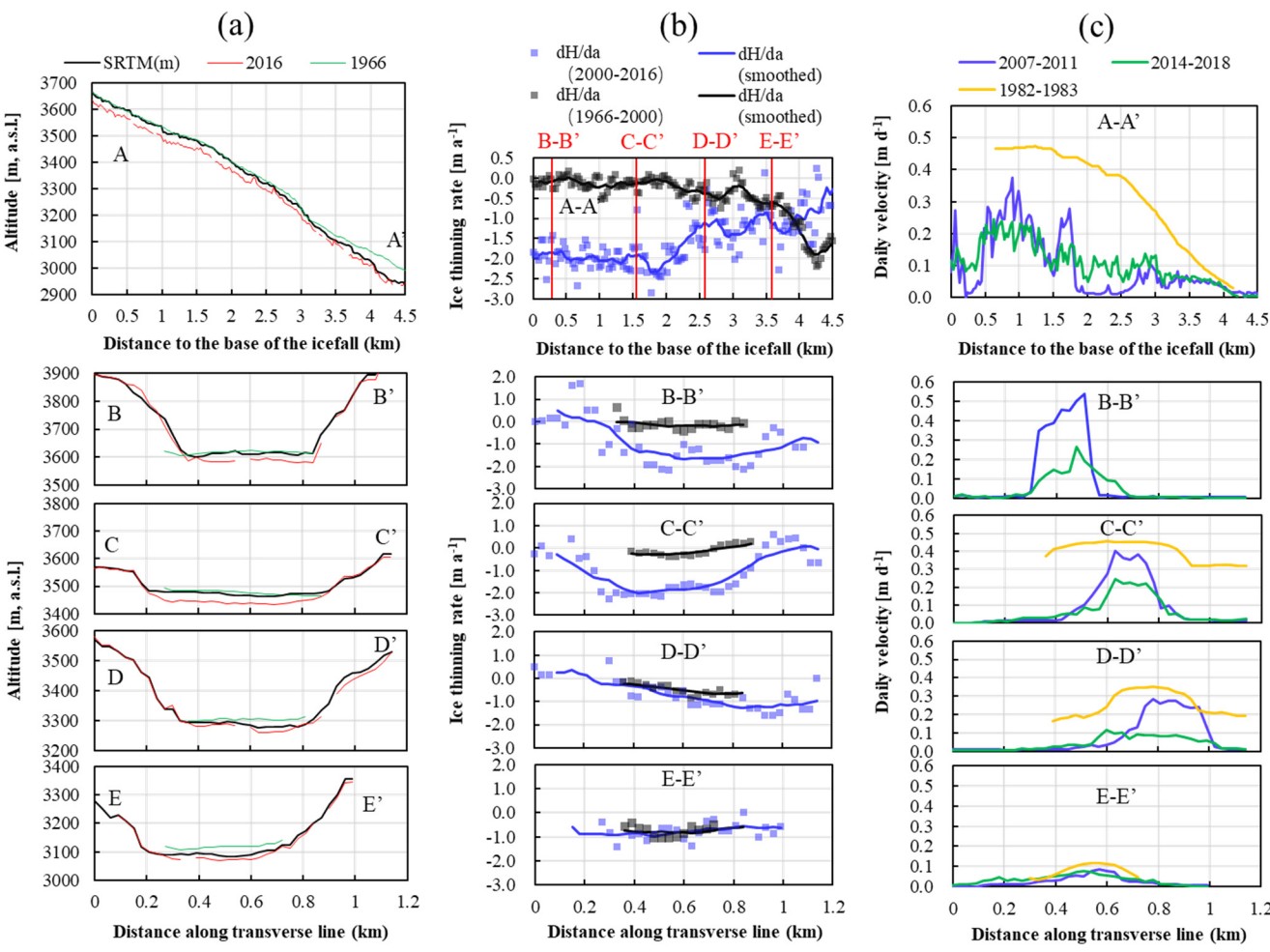

**Figure 3: Comparison of ice surface elevations, average annual ice thinning rates (Brun et al., 2017) and velocity changes (Liu et al., 2019a; Zhang et al., 2010) along profiles A-A' to E-E' in Figure 1.**

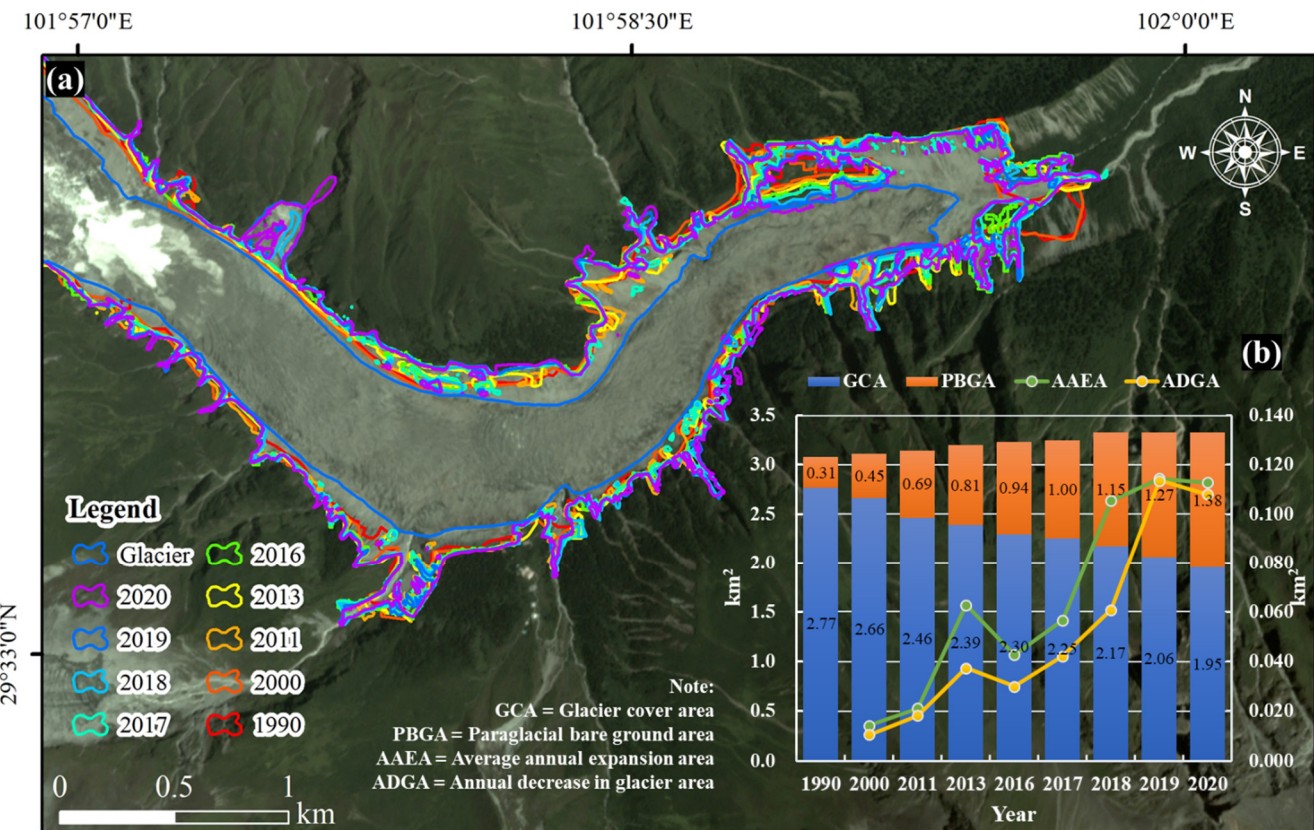

**Figure 4: (a) Mapping results of PSFs in the HLG between 1990 and 2020. The glacier boundary (blue line) is mapped by a**
 **PlanetScope image acquired on August 16, 2019 (background). (b) Changes of the glacier-cover area (GCA, blue) and the paraglacial bare ground area (PBGA, orange) between 1990 and 2020. The green line indicates the average annual expansion area (AAEA) of PBGA. The yellow line indicates the annual decrease in glacier area (ADGA).**

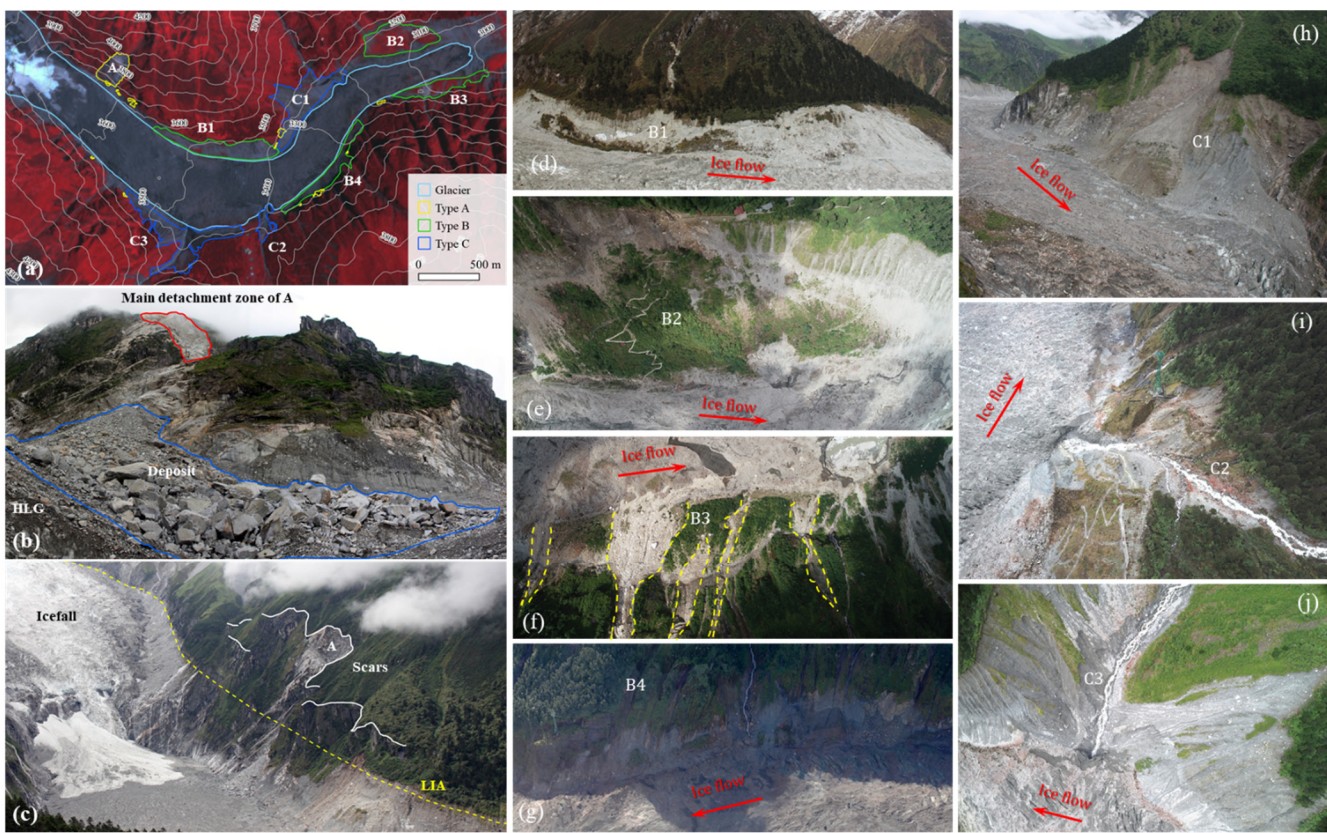

Figure 5: (a) The distribution of three type of PSFs based on Landsat image. (b) The particulars of type A failures in the upper part of the HLG's ice tongue. Main detachment zone (red), Deposit (blue). (c) The scars (white lines) surrounding the rockfall A. Subplots (d, e, f, and g) and (h, i and j) are closer photographs of Type B and Type C failures, respectively, field photo by Qiao Liu.

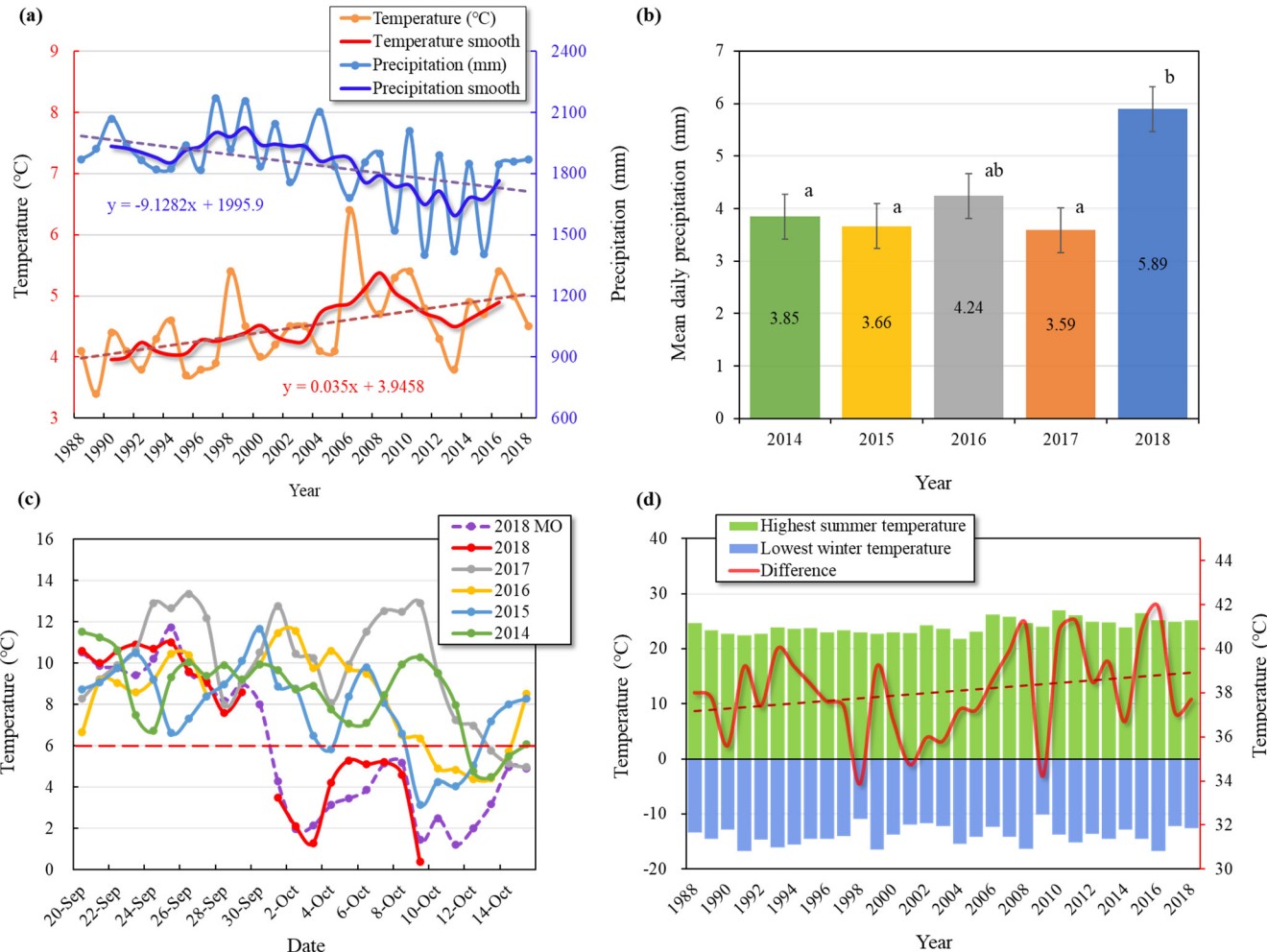

Figure 6: Meteorological data. (a) Mean annual air temperature (orange) and annual precipitation (blue) were recorded at the 3000 m station. The temperature data for 2017 are from manual observations. Between 1990 and 2016, the moving average for temperature and precipitation has been calculated. (b) Differences in the mean daily precipitation between 5-20 Oct 2018 and 5-20 Oct 2014-2017. Analysis of Variance (ANOVA) was used to investigate the difference of mean daily precipitation between five years, it found that mean daily precipitation in 2018 was significantly higher than that in 2014, 2015, and 2017 ($p<0.05$). (c) Differences in the daily mean temperature between September to October 2018 and September to October 2014-2017, MO means manual observation. (d) The average of the highest summer temperature (green bar) and lowest winter temperature (blue bar) of each year, and their differences (red line).

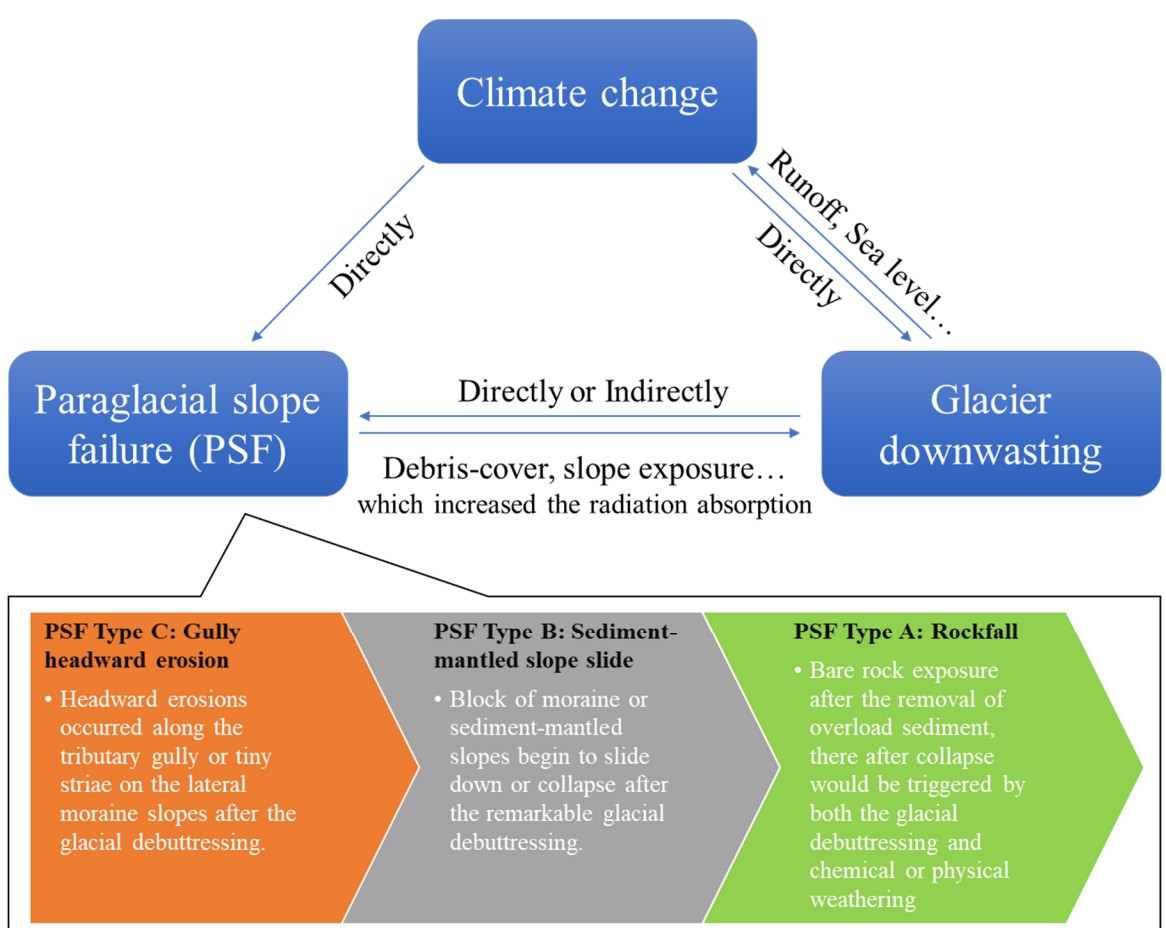

**Figure 7: The relationship among PSF, ongoing climate change, and glacier downwasting. Climate change can directly trigger the PSF through extreme precipitation and sudden increase/decrease of temperature, or indirectly trigger the PSF by glacier downwasting, which can also be used as a preparatory factor to indirectly trigger PSF (e.g., rockfalls). At the same time, many debris fell into the glacier surface, increasing the range and thickness of the supraglacial debris-cover, which in turn affected the surface energy balance for melting and therefore downwasting rate of the glacier, which will thus influence the rate of runoff generation and its contribution to the sea level rise from glacierised catchments.**

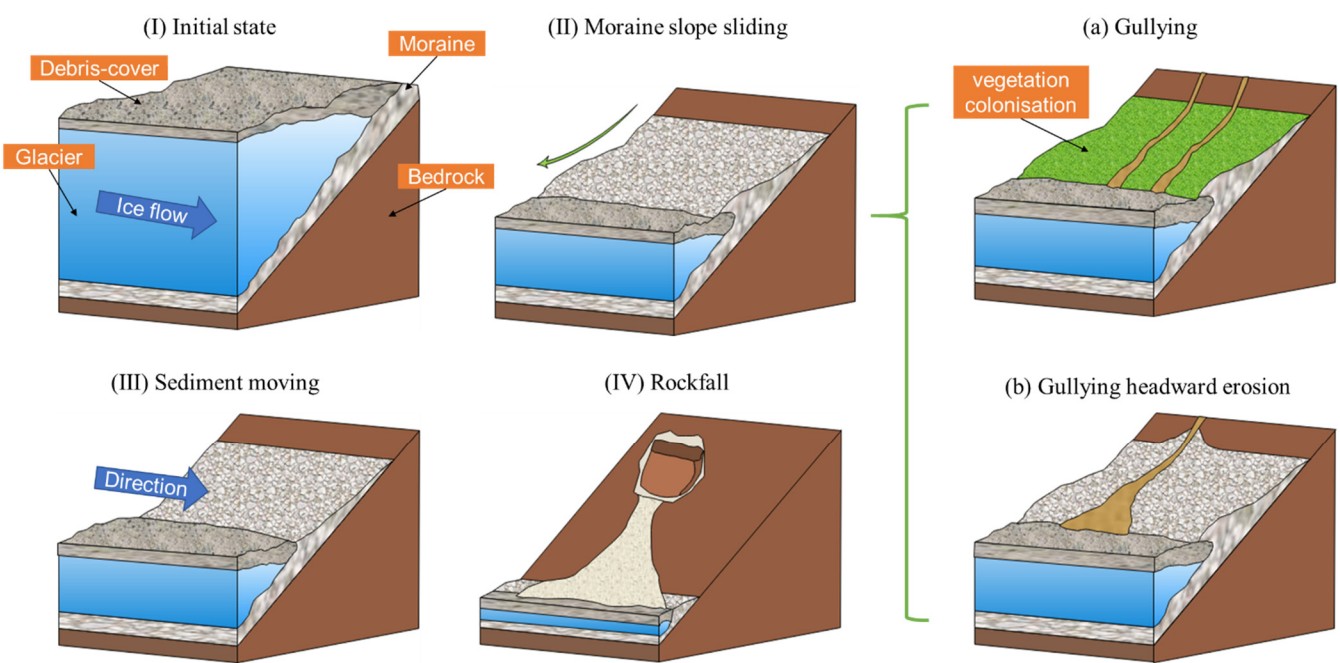

**Figure 8: A conjectural model of paraglacial slope evolution and sediment delivery.**

**Table 1: Satellite/UAV data used to map the PSFs extents**

| Sensor Type | Acquiring Date (YYYY.MM.DD) | Near Infrared (NIR) | Red (R) | Spatial Resolution (m) |
|---|---|---|---|---|
| Landsat TM | 1990.07.08/2000.08.20 | Band4 | Band3 | 30 |
| Sentinel 2 | 2016.05.05/2018.08.23 | Band8 | Band4 | 10 |
| RapidEye | 2011.08.29/2013.05.26/2014.04.16/2015.04.03 | Band5 | Band3 | 5 |
| PlanetScope | 2017.07.17/2019.08.16/2020.08.20 | Band4 | Band3 | 3.125 |
| Google Earth | 2002.10.12 (SPOT5) | Band3 | Band2 | 2.5 |
| DJ-UAV | 2016.08.31/2017.06.07/2018.08.19/2019.05.15 | - | - | 0.1 |

880

**Table 2: Dimensions of PSFs in the study area. We selected periods of high-resolution images with the most obvious terrain changes, the PlanetScope image (2019) for Type A, the UAV images (2016-2019, Fig. S1) for Type B, the PlanetScope, RapidEye, and Google Earth images (2002, 2011, 2014, 2015, 2019) for Type C (locations are shown in Fig. 5a)**

| PSFs type ID | Length (m) | Width (m) | Area (m$^2$) | Displacement speed / Upper edge retreat rate (cm d$^{-1}$) | Increase in exposed area (m$^2$) |
|---|---|---|---|---|---|
| A | 200 | 283 | 47,000 | - | - |
| B1 | 1035 | 114 | 112,424 | 1.96±0.04/- | 12,125 (2016-2019) |
| B2 | 644 | 262 | 122,738 | 2.63±0.04/- | 7,414 (2016-2019) |
| B3 | 805 | 132 | 63,241 | 1.65±0.04/- | 8,528 (2016-2019) |
| B4 | 993 | 103 | 73,270 | 1.66±0.04/- | 10,130 (2016-2019) |
| C1 | 849 | 312 | 139,135 | -/3.39±0.20 | 45,499 (2011-2019) |
| C2 | 164 | 269 | 24,248 | -/0.76±0.11 | 11,923 (2002-2019) |
| C3 | 1097 | 512 | 160,474 | -/1.15±0.15 | 50,989 (2002-2019) |

885