# Peer review of "Intensified paraglacial slope failures due to accelerating downwasting of a temperate glacier in Mt. Gongga, southeastern Tibet Plateau"

_Earth Surface Dynamics, 2021_

## Author Comment (AC3)

**Response to 'Comment on esurf-2021-18' from Jan Henrik Blöthe**

We thank Dr. Blöthe for his very constructive and helpful comments which will clearly help to improve our manuscript by pointing out weaknesses, but also suggesting the respective improvements which we will be necessary to be included for the next version of the manuscript. We try our best to provide as many details as possible to better reply to the reviewer. Please find all the details below.

Below are our responses to the reviewer's comments, with their initial comments in black, our responses in blue, and quotes from the manuscript italicized.

[1] Overall, the manuscript is well written, but sadly lacks rigorous identification of previous work and the work conducted here. Especially in chapter 3 (Methods and data), the authors need to put more effort into making it very clear, which information has been obtained from earlier studies and how the data produced in the framework of this manuscript was produced.

We thank you for pointing out the disadvantage of our manuscript and we will make it clear in the next version.

[2] A very basic and important point that the authors fail to include in their work is a thorough error assessment of all measurements they conducted. This is neither addressed in the methods section, where it remains rather unclear how exactly mapping has been conducted, nor are the errors associated with mapping (based on visual interpretation?) in remotely sensed imagery discussed later in the text. For

Sorry, we did not calculate the error on the visual interpretation and bare ground automatic extraction. Such errors are usually estimated by ±0.5 pixel uncertainty along the boundary shape, we will add the error estimation in the next version of the manuscript.

[3] Chapter 3.2: Here the authors mainly present details of earlier studies that quantified glacier mass balances changes of the HLG. I would recommend to include these background information into the description of the study area (Chapter 2) and shift the focus of Chapter 3.2 to the method applied here.

Thank you for your advice, we will move that background information to Chapter 2 or others.

[4] From the technical description in L158-164, I take that the authors used three digital elevation models (DEMs), namely "the TopoDEM (1966), the Shuttle Radar Topography Mission (SRTM) 30m DEM (2000) and the ASTER-DEM difference between 2000 and 2016". While I am familiar with the latter two (the ASTER DEM data is from Brun et al. 2017, include reference here), I am not aware of the TopoDEM (1966). If this refers to the DEMs calculated in Cao et al. (2019), this needs to be indicated here. Moreover, from the technical details outlined here, it remains unclear to me whether the authors only analyzed five profile lines, or calculated full DEMs of difference and used five profile lines to visualize the data. Please elaborate this in more detail and outline, in the case data was only analyzed along five profile lines, how these are encompassing the full variability of surface changes on the glacier tongue.

TopoDEM (1966) is not calculated by Cao et al. (2019), but has been reported by Liu et al. (2010)

and Zhang et al. (2010). We calculated full DEMs and used five profile lines to visualize the data. We will add more details to make this information clearer.

These profile lines were selected to try to cover every type of slope and keep a certain distance between the transverse profiles so that the extracted results are better for comparison. To present more surface changes information, we will add the dh/dt maps in the supplementary files in the revised manuscript.

[5] Furthermore, in L165-174 (still Chapter 3.2), the results of earlier studies that quantified ice flow dynamics are presented. In the final sentence the authors describe a comparison of "long-term ice flow velocity changes […], based on three-periods results of 1982-1983 (in situ observed), 2007-2011 and 2014-2018 (SAR satellite derived)." Where does this data come from? Is this an analysis done by the authors, or does it refer to the studies presented before? As the dates mentioned here do not match the time spans given for the studies cited above, I would ask the authors to either clearly indicate the provenance of these data.

The *1982-1983 (in situ observed)* data are from L165-168 '*the earliest observations of surface velocity of the HLG were during 1982-1991*'. Here we used the data from the earliest period (1982-1983).

The *2007-2011 and 2014-2018 (SAR satellite derived)* data are from L170-171 '*acquired by PALSAR-1/2 satellites from 2007 to 2018, Liu et al. (2019a)*'. Here we used the data from the two period (2007-2011 and 2014-2018).

We have marked the source of the data in parentheses, if this is still not clear enough, we will consider adding a scanned picture of the 1982-1983 published velocity map and the gridded velocity fields shared by Liu et al. (2019a) in the supplementary files in the revised manuscript.

[6] Chapter 3.3: This is a very brief description of how slope movement was quantified, given that large parts of the results and discussion build upon this data. While manual tracking of tie points in repeated imagery can be considered a fairly robust technique, I would recommend that the authors at least try to quantify the error associated with this tracking of tie points. This can be easily achieved by tracking stable surfaces in the vicinity of the slope failures. Furthermore, may I suggest to include the individual vectors for manually tracked tie points in Fig. S2? Last but not least, let me point out that there are multiple software solutions that allow for automated tracking in consecutive imagery, which would result in full 2D velocity fields that might enable a much deeper insight into the mechanisms of the failures (and reveal local variability).

This is a good suggestion, and we will try to quantify the error in this part in the next version of the manuscript. We will consider including the individual vectors for manually tracked tie points in Fig. S2 if necessary.

Indeed, there are much software that can automatically track in consecutive imagery, and we have tried it. However, because the vegetation on the Type B slope is well-established and fewer tie points can be identified, the automatic tracking effect is not good, we therefore chose manual tracking with higher accuracy.

[7] Section 5.1.1: Here the authors discuss the possible preparatory and triggering factors for the

rock fall they observe during their study period. The authors might want to further elaborate how the exceedance of a precipitation threshold of 60 mm for a single day in June 2018 is connected to the triggering of a rock fall in October 2018. This is mainly referring to L362-65 and Tab. 3, where the authors argue for a precipitation intensity anomaly, which I find hard to follow given the data. Yes, 2018 has seen one day with more than 60 mm of daily rainfall, i.e. 61 mm (L364). Without giving more details on the exact precipitation values for 2016 that also saw three days with more than 40 mm per day, I do not think this can be seen as an indication for a precipitation intensity anomaly. In my view, it would be worth to look at the antecedent rainfall in the five to ten days before the failure and compare antecedent rainfall statistics between years, especially in a setting with very few days without rainfall. Furthermore, in section 5.1.1 the authors argue for a potential triggering by frost action. It is my feeling that also this remains speculative, as the cold interval the authors refer to here (01-09 October 2018; L369, Fig. S4) happened at least 5 days before the failure that the authors date to 15 October 2018. However, the temperature data for 2018 is unavailable for the days following 09 October, making this link questionable.

Thank you for your comment! We can understand your opinion that the precipitation in 2018 is not considered abnormal. We will revise the description of this part in the new version of the manuscript. And we will revise the study time to five to ten days before the rockfall occurred (**5-20 Oct 2018**) and will add the below figure in the manuscript.

[Figure]

The mean daily precipitation between **5 and 20 Oct 2018** is 5.89 mm which is significantly higher than the 2014, 2015 and 2017, although not significant compared with 2016, the number is still high.

As for the temperature data after 9th October 2018, we obtained and checked the daily temperature data of four periods (2 am, 8 am, 2 pm, 8 pm) by manually observation at the 3000m observation station at Mt. Gongga, and calculated and plotted the daily mean temperature curve, which has been added to Fig. S4 (the purple dash line in the below figure). We will revise it in the manuscript.

[Figure]

Note: MO means manual observation.

[8] L35-40: Here the authors gather five modes of response to slope failure, though I have the feeling that mode 5 "paraglacial debris cones and valley fills" is rather the results, i.e. deposit of the processes listed in 1-4.

We thank you for pointing out this error and we will correct it in the next version.

[9] L67-68: Not really relevant at this point, as the tourism activity at the site is detailed in L109-117.

We will remove it.

[10] L101-02: In L152-53, this information is from Zhang et al. 2010, please add reference here

Thank you, we will add it.

[11] L124: How was the glacial area mapped exactly? As the HLG is debris-covered, a precise delineation between debris-covered ice and the debris-covered surroundings is not trivial. Please elaborate in detail, how mapping of glacier extent was conducted and what the associated uncertainty of this mapping was.

Due to the complex terrain of HLG ice surface-numerous ice cliffs and crevasses- as well as the high slope on both sides of the glacier, it is very difficult to use GPS to measure the glacier boundary on the ground. Therefore, at present, the use of medium- and high-resolution remote sensing images has become the most effective method of glacier boundary extraction. Although the boundary of the debris covered glacier is not as distinct as the clean glacier, it is still visible. As the mentioned above comment [2] the errors are usually estimated by ±0.5 pixels.

The following figure takes PL image as an example:

[Figure]

[Figure]

[12] L124-26: Also for the paraglacial slope failures (PSFs), it is not clear how exactly these were mapped. In L121-24 it is described that using the NDVI, vegetation covered areas were excluded. But how exactly was the mapping of the PSFs achieved in remote sensing imagery. What were the criteria for mapping? Was this mapping field-evidence based and if so, when was field-work conducted?

[13] L130-34: Again, it remains unclear at this point how PSF boundaries were extracted, what validation means in this respect and based on which criteria manual correction was done.

As mentioned before, the current method does have the disadvantages you indicate, but it can delineate the expansion of exposed slopes in most area. We will fully discuss the uncertainty of the delineation boundary in the next version of the manuscript. Manual correction of PSF is mainly based on the visual interpretation of the RS image, eliminating the pattern and obvious mis-extraction (relying on empirical judgment).

[14] L138: I would suggest to state this in more detail. At this point, it is unclear, whether the "mean quality of 0.01 m" refers to the position accuracy of the RTK UAV, or to the SFM output. Furthermore, "+1 ppm (RMS) in XY" is neither clear in this regard. In order to allow the reader to follow and to judge the quality of the data used in this study, I suggest to explain in detail, how the UAV images were processed and what the residual mismatch of their geolocation is.

The GNSS quality information are required from DJI official website (https://www.dji.com/uk/phantom-4-rtk/info#specs)

*Horizontal 1 cm + 1 ppm (RMS)*
*1 ppm means the error has a 1mm increase for every 1 km of movement from the aircraft* (we will add this sentence in the next version)

UAV measurement is a very mature technology, albeit a new one; we will describe our UAV mappings in the revised manuscript with some more details.

[15] L142-43: I take it that the authors co-registered and orthorectified the UAV-derived orthomosaics with the PALSAR DEM, or is the sentence correct that individual UAV images were used for this? May I ask the authors to elaborate this a bit more, as this is cionfucing? IN line XX you write that these are already orthorectified?

We use PL images and ALOS PALSAR DEM to co-register and orthorectify the UAV images which are synthesized by ContextCapture Center Master Software based on UAV photographs. Due to the

steep slope of HLG, DEM correction is required for the synthesized images. We will add more detail in this sentence, thank you.

[16] L156: The authors compiled a data set here, but I fail to see where this data set is included in the manuscript? Is there a figure or table that shows this compilation?

We used the dataset in Figure 3 for ice thinning (elevation change) analysis. Both L156-160 illustrate the data used in Figure 3, and we will add some details to make it clear.

[17] L174: In the very brief section following this heading, the "outline change rate" is not mentioned nor explained how this is quantified. Instead, the authors quantify the rate of headscarp erosion. Consider rephrasing the heading here.

Ok, thanks for mention it, we will change the heading to "Slope movement and headscarp erosion rate".

[18] L178-180: May I suggest to elaborate more clearly how the outlines of failures were used to calculate a mean annual retreat rate for headscarps?

We calculated the mean distance between the two outlines where the headscarp erosion occurred at the head of the slope. It can be obtained by using the Average Nearest Neighbor tool in ArcGIS. We will add this sentence in the next version.

[19] L220-23: The time spans given here in the text are not the same as in Fig. 4. What is the rate between 2000 and 2019?

What I want to express here is that the growth rate of EBGA has increased from ~0.01 $km^2$ $a^{-1}$ at the beginning (1990-2000) to 0.1 $km^2$ $a^{-1}$ now (2019-2020).

We will change the time period to every ten years for comparison.[20] L224: Which area are you referring to here?

Sorry to confuse you, but what I want to say here is the area of EBGA.

[21] L225-26: Where is the data for the lower frequency of PSFs?

Sorry, we will add it in the next version.

[22] L227: Where does the knowledge of slope material come from? Did you map out slope material distribution during field work?

We will remove this. The classification of the three typical styles of paraglacial slope failures are mostly based on geomorphological characters. And sorry, we didn't map out the slope material distribution.

[23] L239-44: This description of the rock fall (PSF type A) needs to be refined. In L240 it is stated that the rock fall occurred on a south-west facing slope, while in L244 it is stated that the mass detached from a steep north-facing slope? Is it that the general topography is south-facing and the nice of the detachment faces north? The picture in Fig. 5b, however, does look like the source is also facing the glacier – please clarify.

Sorry, we will revise it. It this paper, we uniformly use the south- and north-facing slope to express

the position of the slope.

[24] L241-42: This is not precise enough. How can a deposit of a rock fall be "450m in height fro the glacier surface" and at the same time, "cover a height of 380 m"?

We will revise the "deposit" to "rockfall area" or others.

We will change the sentence to "*The rockfall area is 283 m in width, 200 m in length, covers a 2D area of ~47,000 m², with a vertical height of 380 m and a slope length of 472 m.*"

[25] L258-59: Please indicate what magnitude is considered small here, as to me "each with a mean area of 750 m2" is not clear.

All the small rockfalls found were all smaller than 4,000 m², with a mean area of 750 m², much smaller than the large rockfalls (47,000 m²) that occurred in 2018. We will describe it more clear in the revision.

[26] L262-63: Please try to be consistent in labelling the processes: "Sediment-mantled slopes slide and collapse" vs. "Sediment-mantled slope slide and collapse". In my view, both seem a bit clumsy – you might want to rephrase.

Thank you for pointing out that, we will be consistent in labelling the processes: "*Sediment-mantled slopes slide and collapse*". "*Sediment-mantled slopes*" is a term in paraglacial geomorphology, please see "Colin K. Ballantyne, Paraglacial geomorphology, Quaternary Science Reviews, Volume 21, Issues 18–19, 2002, https://doi.org/10.1016/S0277-3791(02)00005-7"

[27] L263-66: The numbers given here are surprisingly round and do not match the sums of the individual numbers mentioned in Tab. 2. Also, in Tab. 2, there are no errors associated with the numbers given for the area. Please include a statement on the precision of these estimates in the text. This also applies to L310.

The four Type B slope areas in Table 2 add up to 371673m², which is approximately 370,000 in L264. We will add "~" in the next version.

As for the number "297,000 m²" in L265 is the total increase area from 1990 to 2020 calculated though satellite images, which is not the one calculated from UAV imagery (2016-2019).

We will add more details about the use of the data in Table 2 so that audiences can understand. Thank you for pointing out.

[28] L271-75: This comes as a surprise here and rather belongs to the discussion.

Thanks for mention, we will correct it.

[29] L276-78: How was the error of 0.04 cm d-1 quantified?

From L149 "*mean quality of 0.15 m in XY*", error automatically calculated by ground control points during the UAV registration procedure.

[30] L283: How do the authors know that the detectable slope movement began around 2000? Has this been published, or did the authors run additional analysis beyond the 2016-2019 UAV surveys that have been described in the text. Same applies for L296.

According to the historical remote sensing images (2000 TM, 2002 Google earth, 2011, 2013 RapidEye, please see the below figure), the deformation of the slope began to appear in 2000. Same applies for L296.

[Figure]

[31] L286: "the landslide has fallen" sounds as if it was a vertical movement that was quantified? Until now, it is my understanding that the authors quantified 2D horizontal displacement.

Yes, the landslide was quantified only in 2D horizontal directions. The slide rate could be estimated based on the surface slope of the terrain. Thanks for mention, we will correct the wording.

[32] L400-02: This comes as a surprise as a) in L361-62 the authors argue that 2018 has seen relatively low temperature. Furthermore, Fig. 9 suggests that 2017 has a data gap in temperature readings.

Thanks for mention, we will correct it.

[33] L430-38: I am not sure how this is connected to the data and topic presented in this study? Consider removing paragraph or elaborate the connection to paraglacial slope adjustment.

Thank you for your suggestion! We decided to add more detail to make the link between glacier velocity and hillslope processes clearer.

The slowing of the glacier velocity and the glacier thinning corroborate each other. On the one hand, the thinning of the glacier slows down the glacier velocity; on the other hand, when the glacier velocity slows down, the ice flux transported from upstream to downstream is reduced, which accelerates the glacier thinning and ultimately leads to the slope sliding.

[34] L444-46: Is this supposed to be a general statement? Also, check grammar.

Thanks for mention, we will correct it.

[35] L448-50: It is hard to see how the deposition of debris onto the glacier is directly affecting climate. As the authors try to outline in Figure 8, debris cover generated by slope failure might have

an effect on glacier downwasting, which in turn can have a tiny effect on the climate.

Yes, as show in Figure 8, debris cover generated by slope failure might have an indirectly effect the climate.

L456: We will change the sentence to "…which in turn affected the surface energy balance for melting and therefore downwasting rate of the glacier, which will thus influence the rate of runoff generation and its contribution to the sea level rise from glacierised catchments."

[36] L464-70: While McColl and Davies (2013) showed that also failures of similar magnitude as B2 in the present study can deform glacier-ice at the rate of mm/yr (assuming a min. average thickness of ~1 m), the evidence presented here is limited and the discussion of this aspect remains too surficial. It might be a way forward, to present the displacement data in more detail and quantify the supposed narrowing and squeezing effect that is mentioned here to back this aspect with data.

As reply in comment [6] above, we will include the individual vectors for manually tracked tie points in Fig.S2 according to your suggestion.

[37] L474-75: What data is this statement based on? Is there a study that found this increase in debris-flows and flash-floods that could be cited here?

Yes, there is a study can be cited here: "Lu, R., and Gao, S. (1992). Debris flow in the ice tongue area of Hailuogou Glacier on the Eastern slope of Mt. Gongga. Journal of Glaciology and Geocryology, 73-80". However, there is no data and literature to support the increase in frequency of debris-flows and flashfloods, we will remove "*increased frequency of*", and cite the literature in the text, thanks for mention.

[38] L482: In L195-96 the authors state that between 2016-2019 the glacier terminus retreated by >150 m with a rate of ~52 m yr-1. Which is true?

I'm sorry, it's 450 m (15 m $a^{-1}$) in L482. We will check all the numbers in the next version.

[40] L485-86: In your data, type A landslides are limited to one rock fall. Did you find evidence for debris avalanches as well?

No, we didn't find any debris avalanches.

[41] Figure 1: Might I suggest to add the glacier outline for the years 1982/83, as in the text the authors have quantified (or cited) the displacement values for this period (Fig. 3).

Ok, we will add it.

[42] Figure 3: Is there a reason for the black line being thinner in D-D` of the first column? Also, in the caption, replace annual thinning rate with average annual thinning rate, as these have been quantified over multiple years, right? What remains unclear is the provenance of the data shown in the right column. Has this data been calculated from remote sensing data by the authors? If not, add the references to the data to the caption.

Thank you for your careful review. We have corrected the D-D 'line as shown in the following picture and will correct the caption and add the reference.

[Figure]

[43] Figure 4: Change "(white line)" to "(blue line)" in the caption, as this is showing the glacier outline, judging from the legend.

Thanks for mention, we will correct it.

[44] Figure 6: Might I suggest to label the vertical axis of the last column with "displacement velocity"? Slide velocity implies a vertical component, but these are horizontal displacement values, right? Furthermore, in the caption, what does the reference to Qiao Liu et al. imply? Has this data been published before? I cannot find this reference in the reference list.

Thanks for mention, we will correct the vertical axis to "displacement velocity". This data has not been published, it was requested by the ESD journal to include the source in the maps and photos. The UAV images were taken by Qiao (Corresponding author of this manuscript) et al and therefore need to be noted here.

[45] Figure 8: May I suggest to give this figure a more detailed caption that explains the arrows and the reciprocal effects? Also, why does Type C have a larger arrow than Type A and B, what exactly does "remarkable glacial debuttressing" imply? Also check grammar of text box PSF Type A.

Ok, we will add more detail on the caption. The three arrows of type A-C on the right end are just for the sake of aesthetics, nothing else.

[46] Figure 9: Here it should be indicated over which time window the moving average for temperature and precipitation has been calculated. Furthermore, the reason for and length of the temperature data gap in 2017 should be mentioned in the caption. Also, are Tem_smooth and Pre_smooth appropriate abbreviations for a running mean of these variables?

Thanks for mention, we will correct it.

---

## Author Response (AR1)

**The author's response on esurf-2021-18**

Dear editor Krautblatter and two referees,

Thank you for taking the time to review our manuscript. We have carefully addressed all the comments made by the two referees, Samuel McColl and Jan Henrik Blöthe. Their constructive suggestions resulted in numerous changes throughout the manuscript. We revised the key research question and the objectives, detailed the data and methods, reconstructed the discussion to make better use of our data and results, and we think we have greatly improved it.

In the initial AC replies (AC2 and AC3), we have responded to the reviewers' comments point by point. This document added to the location of these changes in the manuscript (without track changes) and includes our new responses to some comments.

With kind regards,
On behalf of all authors
Yan Zhong and Qiao Liu

**Response to 'Comment on esurf-2021-18' from Samuel McColl**

We thank Dr. McColl for his very constructive and detailed review which will clearly help to improve this paper by pointing out weaknesses but also suggesting the respective improvements which will be necessary to be included in the next version of the manuscript. We try our best to provide as many details as possible to better reply to the reviewer. Please find all the details below.

This paper documents changes to the unstable hillslopes adjacent to a retreating glacier using observations from satellite and UAV imagery and field visits. Three main types of hillslope response are documented – rockfall, debris sliding, and gullying – and the authors attempt to explain the occurrence of these with reference to observed changes in the climate and glacier. The paper presents data that is of interest for several reasons; 1) the research context - a relatively understudied type of glacier environment [monsoon temperate, within Southeast Tibet]; 2) several types of hillslope failure mechanisms are documented, adding to the growing knowledge of the complex response of hillslopes to glacier retreat; 3) the authors use these data to make observations on the interactions between hillslope and glacier processes, topical for research on alpine hazards, climate science, and paraglacial geomorphology. While the observations are of interest, the overall purpose and aim of the manuscript is a little unclear from the introduction, without the identification of key research questions tied to the objectives or without a set of hypotheses being developed and then tested. It rather feels like the data was collected for the sake of collecting it in the hope that it might yield some interesting observations (which it does, but the lack of a clear study setup/purpose detracts from the paper's impact). The manuscript falls short of delivering any major discoveries that change the current understanding of landscape response to deglaciation, instead providing a more incremental increase in data documenting the range of responses (which in itself can be useful, but is currently not sufficiently capitalised upon). Other than further confirming the significant role of glacier down-wasting (which is already a very well-established process of hillslope destabilisation), the manuscript is unable to provide much more than speculation on other factors responsible for the hillslope response patterns observed (e.g. long-term strength reduction or frost weathering responsible for the rockfalls observed). On the interactions between the hillslopes and the glacier, these are also rather descriptive without detailed analysis. In my 'specific comments' below I offer some suggestions for how you might address some of these short comings and better utilise your (nice) data to give your manuscript more impact. I hope that the specific comments and technical corrections below will be useful in reshaping this manuscript and enhancing its clarity, focus, and overall contribution.

While a growing body of research focusing on process of paraglacial hillslopes destabilization of the European Alpine, Southern Alps, and Cordillera Blanca (Andes), the representation of evidential findings in the southeast Tibetan Plateau (low latitude region) is still unknown. We speculated that the recent high frequency of glacial debris flow in SE Tibet may relate to the numerous paraglacial slope failures, which act as debris source of the strongly deglaciated catchments during precipitation rich seasons in this region. To test this hypothesis, we decide to select a temperate glacier for a detailed study of glacier-slope interactions in SE Tibet. However, due to the topographical restrictive, sparsely populated, and cloud-cover, most of the glaciers in SE Tibet are difficult to be observed on the ground and by satellites for the long-term.

Hailuogou Glacier (HLG) is one of the largest temperate glaciers in Mt. Gongga which has the lowest elevation in its ice tongue area. Due to its high visibility and accessibility, it has been well documented and observed since the early 20th century. Based on these data, we could possibly relate the historical and recent glacier changes with the paraglacial geomorphology responds observed during the satellite era as well as in the resent UAV epoch. Studying this process is also of great significance for understanding the causes of frequent glacier-related hazards in SE Tibet. Meanwhile, the HLG is an important freshwater and tourism resource in the east Mt. Gongga. Various geological hazards caused by the hillslope instability have an important impact on the socio-economic development of the downstream. Analysis of existing data is therefore particularly urgently needed.

We agree with your comment that our current study is still descriptive since we had not monitored or modelled the physical mechanics processes of the slope, rock and ice, thus the long-term strength reduction or frost weathering responsible for the rockfalls observed yet could not be quantitatively examined. Thank you for listing suggestions below based on which we hope we could reshape the manuscript by analysing the interactions between the hillslopes and the glacier in more detail and strengthening our contribution by making better use of observed data.
* * *
[1] The introduction needs to do a better job of setting up the objectives of this work. For example, what key research gaps or questions (e.g. in paraglacial hillslope response) is the manuscript addressing? How will your 3 objectives stated on L82-87 help to address these? Why is the HLG study site important/useful for addressing these questions? (I note that on L113-117 there is clearly a hazard motivation for the selection of the HLG site, so perhaps this could be presented better in the introduction as one of the justifications for addressing the key research questions and for the selection of the HLG site).

We added paraglacial hillslope response in SE Tibetan as the key research gap and other information in the introduction. We also revised our objectives to make them clearer (L71-86).

[2] The classification of the types of hillslope response could be better justified and used consistently throughout the manuscript: e.g., the description of the three types (A, B, C) are described differently in the abstract than in the conclusion (in which they are also labelled differently, as i, ii, iii). On lines 35-40 five main classes/modes of hillslope response are described from the literature, but it is not clear how the three types (Type A, B, C) described in the manuscript relate to these 5 modes. Also note that Mode (5) paraglacial debris cones and valley fills seems to describe a product of hillslope erosion, not a type/process/mechanism of hillslope response. Debris cones and valley fills presumably can be produced by a variety of mass movement processes. So there is some inconsistency in the 5 modes presented.

We thank you for pointing out this error and we moved the Mode (5) paraglacial debris cones and valley fills in the new version, and we have changed the label in the conclusion section.

[3] The term Paraglacial Slope Failures (PSFs) is used as a catch-all for the three main hillslope responses that are documented. However, given that the Type C response seems to be focused on headward gully erosion and said to involve fluvial processes (L411), it casts doubt on the suitability of the term 'Paraglacial Slope Failure' to represent this, as fluvial processes are not traditionally

considered to be a slope failure process. Perhaps a better term is needed?

[4] Related to the previous comment, I feel that the processes involved in the Type C response are somewhat unclear. Does this response type involve mostly debris flow processes (in which case the term PSF may be OK), or is it mostly fluvial erosion (rilling and gullying)? While I suspect there are not sufficient (temporal) data to confidently identify the processes causing expansion of these gully areas, perhaps there are clues from the deposits they produce (i.e. are the fans below these gullies more typical of fluvial or debris flow process?).

Thank you for your suggestion. As you have mentioned, Type C is an incredibly unique process in HLG even in the whole SE Tibet. Most of them are based on Type B, and their shapes are different from that of Type B's (from arc to triangle) because of the strong gully erosion upstream. In fact, gully erosion also exists in B1-4, but they are not obvious enough to change the overall shape of the unstable slope and are not classified as Type C. So essentially, Type C is also one of the PSF landforms.

[5] For each section of the Data and Methods (i.e. sections 3.1, 3.2, 3.3, 3.4) I suggest starting with a brief explanation of the purpose of the method, trying to link back to the objectives where relevant. This will help readers to understand why you are doing each part.

We agree that and have added more details in the section of the Data and Methods to help readers to understand this part clearer.

[6] I would like to see some more explanation of how the three response types were identified, i.e. what key criteria were considered (i.e. elaborating on the comment on L228-230). As I understand it, they key criterion for assessing the presence of a hillslope response was the appearance of bare ground (especially sediment?). I have two potential issues/queries with this:

This suggestion is very pertinent. To help readers better understand how these three types of slopes are identified and classified, we draw up a table (Table 1) to show the classification reasons and results of each slope (A, B1-4, C1-3) in section *3.1 PSFs mapping and classification*.

Table 1

| Types | | Sub-types | | Classification Standard | Classification Result |
|---|---|---|---|---|---|
| ID | Name | ID | Name | | |
| 1.0 Rock slope failures | | 1.1 | Rock avalanche | Large-scale, catastrophic rock slope failure | - |
| | | 1.2 | Rockfall | Local-scale, high-frequency, and discrete rockfalls | A |
| | | 1.3 | Deep-seated gravitational slope deformations | Extremely slow flows or displacements of bedrock | - |
| 2.0 Sediment slope failures | | 2.1 | Sediment-mantled slopes slide and collapse | Rock mass slides along the shear plane Shape: arc or strip | B1-B4 |
| | | 2.2 | Gulley headward | Strong headward erosion is added | C1-C3 |

a) not all bare ground exposed by the glacier will be unstable, so how do you differentiate unstable ground from stable ground (e.g. using signs of disturbance, or a slope angle threshold?), and can you quantify the abundance of stable vs unstable ground?

Indeed, not all bare ground is unstable, and the slope angle difference between the stable and unstable slopes on both sides of the glacier is different to some extent, but they cannot be used to identify them all. In the process of bare ground extraction by NDVI, we found that some areas showed an interannual change in shape (e.g. Type C), while some areas showed a sudden increase in exposure area for a short time (e.g. Type A), and vegetation developed in some areas with the obvious movement of vegetation patches. (e.g. Type B2-3). These become the important basis for us to identify the unstable slope. The temporal and spatial resolution of the data is both higher than the existing DEM data, so it is the key to obtain an unstable slope boundary under current conditions. Finally, we try to quantify the abundance of stable vs unstable ground of three response types in nine study periods between 1990-2020, please see the table below (Table 2).

Table 2

| Years | Abundance | |
|---|---|---|
| | Unstable ground | Stable ground |
| 1990 | 83% | 17% |
| 2000 | 64% | 36% |
| 2011 | 54% | 46% |
| 2013 | 51% | 49% |
| 2016 | 59% | 41% |
| 2017 | 57% | 43% |
| 2018 | 58% | 42% |
| 2019 | 55% | 45% |
| 2020 | 56% | 44% |

b) You state that due to the climate conditions vegetation colonisation is extremely fast in this environment. This presumably means that much of the bare ground exposed in the early part of your study becomes colonised by vegetation by the end of your study, especially for Type B responses which do not necessarily prevent vegetation from establishing on the main body (i.e. vegetation rafting). Does this pose a challenge for the identification of bare ground through time, and if so how do you get around this or how much does it affect your results?

Indeed, vegetation colonisation is extremely fast in this Mt. Gongga. This also makes the area of bare ground in this environment much smaller than the paraglacial environment in the central Himalayas. However, vegetation colonisation is limited by slope, elevation, and time. In four years of UAV images, we observed that the rapid vegetation colonisation mainly affected the bare ground areas in glacier foreland after the ice tongue has completely retreated. Therefore, we eliminated this

part of the area from the automatically extracted PSFs during classification and their geometry calculation.

We added "*For better inter-annual comparisons, we then manually removed those patches and areas where the PBGA has changed due to vegetation colonisation in some paraglacial slopes and in all glacier foreland to keep the remaining unstable slope areas as final mapped PSFs.*" in L125-127.

[7] L148-149: Please elaborate upon how you calculated the 'mean quality of 0.15 m in XY'. Did you calculate this by withholding a sub-set of your ground control points not used in the georegistration, in order to provide an independent check of georegistration error? It would be preferable to also provide the maximum error or at least a measure of dispersion (e.g. standard deviation), not just the mean. Please explain what you mean by 'successfully occupied positions' or revise this wording to make it clearer what you are referring to.

We used all ground control points in four study periods to calculate the georegistration error and added the maximum error in this section.

We revised this sentence to "*Overall, the final reported quality of the co-registered UAV images were quite satisfactory (with a total and maximum RMS error of 1.22 and 1.98 pixel respectively, i.e. a mean horizontal error of 0.15 m).*" (L153-154)

[8] The five profile lines A-E and their data (ice surface elevation, thinning rates, flow velocity) provide some interesting data on changes to the glacier, but it is hard to see how these data are actually used to help understand the hillslope response processes. It is unfortunate that these data are not more thoroughly used to explore the relationships in space and time between glacier changes and hillslope changes. Although on L452-454 you state 'our analyses show a temporal and spatial component to PSF development' there is no real attempt to combine the data of Figure 3 with the data of Figures 4-7 in any quantitative way. It might have been worth setting up a hypothesis, for example that the rate of Type B movement will correlate with the rate of ice thinning, or that the magnitude of ice thinning would correlate with the growth in size of Type B and Type C responses, and then quantitatively/systematically test these. Such relationships are only somewhat qualitatively/subjectively commented on in the manuscript. Likewise, why were four (and not some other number of) transverse profiles chosen and what was the rationale for their placement – was the hypothesis that thinning and hillslope response will differ depending on distance up-valley of the terminus, or were there differences in terrain type, geology, or some other environmental variable that were being captured in these profiles?

Thank you for your advice, we added/reconstructed a new section *5.2.2 Paraglacial slope process interaction with the dynamics of temperate glacier* to maximize the use of our data.

We added a sentence in L84-185 to explain the four transverse lines chosen.

[9] The glacier velocity data are interesting in themselves but do not seem to be well utilised or particularly relevant to the objectives. What was the hypothesis that was being explored with this, or why would changes in glacier velocity be expected to cause (or respond to?) hillslope processes? On L430-438 we get a sense that the velocity data is being used to infer that the high flow rates have been responsible for a high erosion rate and steepening of the valley flanks. But this is not supported

with any data or further context – there are no data to show that the valley walls are steeper than other glacial valleys with lower flow velocities, and there is no comparison made between the rates of hillslope response in the HLG to other locations to explore whether the rates are unusually high and therefore are correlated with a high flow rate. I suggest that unless there is a good case for retaining the glacier velocity data, then it is removed from the manuscript because currently it adds little insight into the hillslope processes observed.

Thank you for your suggestion! The slowing of the glacier velocity and the glacier thinning corroborate each other. On the one hand, the thinning of the glacier slows down the glacier velocity; on the other hand, when the glacier velocity slows down, the ice flux transported from upstream to downstream is reduced, which accelerates the glacier thinning and ultimately leads to the slope sliding.

[10] There could be more information provided on the rockfalls. At present it seems that only the largest is described in any detail, with the other failures described only in general terms (e.g L253 'We also observe other major rockfalls…suggesting that numerous smaller scale rock falls have occurred in this locality' or L258 'small magnitude rockfalls occur more frequently' Can you present the data for these events – e.g. a freq/mag histogram or table showing their source elevations, when they occurred, and their magnitude, and a map showing location? Presenting these data may help to tease out relationships between the failure patterns and the factors governing them.

Sorry these scars of smaller scale rock falls are different from the small magnitude rockfalls in the below paragraph, we removed the word "smaller" to avoid reader confusion.

[11] The role of mass movements for producing supraglacial debris (e.g. L45-43) seems to be a theme introduced and returned to several times in the manuscript, but at present the manuscript makes little contribution to this topic. While rockfalls were observed to deposit sediment onto the glacier (e.g. L260, L448), this manuscript is hardly the first to identify the role of rockfall in producing supraglacial material so this is not a particularly helpful finding. Moreover the actual effect that these few documented failures have had to glacier ablation is not in anyway quantified in the manuscript, so as it stands the qualitative observation of supraglacial debris accumulation is not particularly insightful. Therefore, I would suggest that either this aspect of the manuscript is removed, to improve the focus of the manuscript, or this aspect is enhanced. Enhancements could be to:

a) provide more quantitative data on the total areal contributions to supraglacial cover of the rockfalls documented in the manuscript and make comparisons with other studies (i.e. substantiate the statement on L448-450 with data and context).

b) include a more detailed description of the contributions (or not) of the other two types of hillslope response (B and C). To what extent have these processes also delivered supraglacial material to the glacier during the observation period, and if they have been delivering sediment then to what extent has supraglacial sediment delivery by these types of hillslope process previously been documented in the literature, and are your findings consistent with that?

c) there is a nice opportunity to discuss sediment delivery to glacier systems more widely than just supraglacial sediment delivery. You pick up on the fact that some of the Type B failures are

deforming the glacier (sensu McColl and Davies 2013), which is a nice observation – to what extent are these failures also delivering sediment subglacially, similar to what was identified by Cody et al., 2020 in the Fox Valley, or are the slopes at your site not engaging this recently-documented sediment pathway? Further, it appears that some of the Type C processes are providing sub-glacial water supply, and therefore presumably these are also delivering sediment to the sub-glacial environment? If so, exploring to what extent these paraglacial transport pathways (i.e. from recently exposed moraines) have been previously documented in the literature would be a good point of discussion.

Thank you for your three very constructive suggestions! We have followed your above suggestions and added the analysis mentioned below in the manuscript.

We calculated the total areal rockfalls covered is 62216 m$^2$ ($10^3$~$10^4$ m$^3$); according to Scherler et al. (2018) the supraglacial debris cover of HLG is 2.71 km$^2$, as a result, the total areal contributions to supraglacial cover of the rockfalls is 2.3%. (L509-516)

As shown in figure 1 (please see below), Type B and C slopes, often have a deeper contact with glaciers during the slow slide and may deliver sediments underneath the glacier. However, there are still a limited number of materials that are delivered to the supraglacial environment through direct rolled (the sediments are more dispersed compared to rockfalls) and through debris flow (faster changes). Type C slopes can also deliver materials to sub-glacial environment through runoff. Their contribution to glacial sediments is hard to quantify, but it is clear that this process does result in a long-term and steady sediments supply to the glacier. As you said that Type B 2 failures are deforming the glacier, and at the same time we also observed the delivery of local sediments (Figure 1, B2). In the 718 days from 2016/8/31 to 2018/8/19, two parts of the slope collapsed while sliding, and the collapsed area increased by 94% in total, with an increase rate of 8 m$^2$ d$^{-1}$. (L516-525)

[Figure]

Figure 1

[12] Related to the previous point, on L45 you refer to the role of 'high-frequency, low magnitude PSFs' in delivering a 'considerable volume' of debris onto glacier surfaces. But what about the low-frequency, high-magnitude events (e.g., large rock avalanches) that are well documented in the literature for their role in dramatically changing glacier ablation? Why do you focus on the small high-frequency events here?

Indeed, rock avalanches are well documented in European Alps, Southern Alps, and others. However, we have not observed such an event at HLG (it may exist, but it has not been documented), and the 2018-event is 'rockfall' in terms of both scale of collapse and impact. We made an explanation in section *5.1.1 Rockfall*.

[13] L265, L270, & L287. Please describe the process(es) by which the Type B features become larger; e.g. is this through glacier downwasting exposing more of the slope, lateral expansion of the failure mass, or headward expansion from retrogressive failure or degradation of the scarp (e.g. from surface erosion processes)?

Both processes exist! We added a sentence in L282-284 to explain.

[14] The observation of nested processes (e.g. gullies developing within Type B failures) is nice but it would have been great to see an analysis of the temporal evolution of these. For example, in Cody et al., 2020 they describe a temporal evolution in hillslope response, whereby moraines initially begin collapsing through sliding and internal deformation, and then later surficial debris flow processes (i.e. gully forming processes) takeover, and eventually both processes relax as the slope adjusts to its angle of repose. Are you able to see a similar or a different evolution in the slopes you observe? This would make another very nice comparison.

This is a useful idea. We observed that the debris flow occurred first, and then the slope slipped. We added the below figure (figure 2) and the text to show their evolution and compare them with Cody's research in L421-428.

Based on 17 years (2002-2019) RS monitoring of the B3 slope, we found that the surficial debris flow occurred before the hillslope movement. Firstly, some surficial debris flows occurred, forming 3 gullies, and no slide was evident in 2002. Secondly, the debris flow gullies gradually expanded, the number increased to 5, the slope slid slightly, and the slide cracks were started to form around in 2013. Finally, the debris flow gullies increased to 6 and expanded further (the largest gully is about 130 m wide at its widest point), the slope slide significantly (up to 2 cm d$^{-1}$ between 2017 and 2018), and the cracks increased in response to rapid glacier downwasting in 2019, which is different from the discovery of the Fox glacier in New Zealand by Cody et al., 2020.

[Figure]

Figure 2

[15] L272-275: Comparison is drawn between the Type B failure process and the conceptual model of moraine evolution by Eichel et al (2018). However, this comparison is hard to follow. Eichel et al describe a transition from an unstable state dominated by debris flows and gully erosion through to a period of solifluction modification, through to stabilisation. It appears to me that the typical Type B hillslope responses you describe in this manuscript seems to be more dominated by debris sliding than debris flow or gullying, and therefore is not a great comparison to stage A of the Eichel model (for the sites they studied) – perhaps your Type C is a better comparison? It would be good therefore, if you could further explain why you make this comparison, and it would be really interesting if you additionally compare the evolution of the moraines at your site to the other two stages of the Eichel model – do they also transition to solifluction and then stabilisation, i.e. the older lateral moraines nearer to the LIA terminus? Perhaps you can identify solifluction features in the imagery data or from your field visits, or perhaps the climate is not suitable for this? If you do find differences then you could instead suggest that at your site you observe a different evolution pathway to what is found for the European Alps? This would be a nice contrast and provide a rich vein of discussion (in the discussion section) if in fact there are differences that can be observed.

Thanks for mention, we have removed it.

[16] L300-306: Please elaborate further on what is meant by a 'transition form', and 'landsliding behaviour' and what is different between the two zones referred to.

Thank you for this comment. B4 is a transitional type between Type B and Type C. From the geomorphological point of view, we divide it into Type B, but it is more connected to Type C than B1-B3, and the proportion of landslides is also smaller (24.9%). *'Exhibits landsliding behavior in two distinct zones'* means 'Landslides can be observed in two zones'. We have corrected the wording

in L317 to improve the clarity of expression.

[17] The observation that south-facing slopes were generally more unstable than northfacing slopes is a potentially interesting observation, but one that is not robustly analysed. For example, the authors might consider a wider range of (intrinsic and extrinsic) factors explaining this difference, e.g:

a) differences in the availability of material (i.e. asymmetrical deposition of glacial drift and moraine construction on either side of the valley), with more sediment on the south-facing slopes;

b) asymmetry in morphology (i.e. differences in slope angle). The latter could be easily tested using DEM analysis; the former could possibly be explored through aerial image interpretation?

Thank you for your constructive suggestion! We have added more analysis based on these two points in L552-559.

[18] Section 5.1. Unfortunately, this section is heavily reliant on speculation, and analysis of only the 2018 rockfall. Analysing the location and timing of a single (and a not particularly spectacular) rockfall in the valley is not sufficient for making meaningful generalisations of the causes of rockfall in the valley. Perhaps this section could be strengthened if more attention was paid to the smaller rockfalls - e. examining patterns in the timing and location of several failures and not just a single failure.

As stated in [10], it is difficult to confirm the specific time of small rockfalls occurred, so we cannot do too much detailed analysis. In addition, the locations of small rockfalls are also very scattered (see Fig 5a and Fig S7). Nevertheless, we increase the discussion and analysis of small rockfalls in the L273, L277, and L363-374 to make the research more rigorous. There are many studies (Huggel et al. 2005; Fischer et al. 2010) of single rockfall/rock avalanche events, and our analysis is more than just speculation.

[19] L389-390: 'instability typically have a slope angle of 25'. Upon what basis is this statement made? Do you systematically measure the slope angles from the DEMs? Are you able to more robustly compare slope angles between the unstable debris-covered slopes and the stable debris-covered slopes to test whether the unstable sites tend to be oversteepened? Perhaps an examination of the slope angle of stabilised moraines closer to the LIA terminus will give some rough indication of the 'long-term' angle of repose of the till making up the moraines in the valley. This could provide a useful test of your hypothesis presented on L457 'we hypothesise that this (moraine collapse) will continue until critical angles of repose are reached which will be followed by vegetation colonization and advanced soil development'.

We did do some slope analysis, and we combine comment [6][17] and this part to make better use of slope analysis from the DEM.

The slope analysis results of 2016 DEM in HLG are shown in the figure 3 (below) which is synthesized with 2000 SRTM DEM and the High Mountain Asia Glacier mass balances from 2000 to 2016 published by Burn et al. Interestingly, we found that the Type B (with a mean slope of 29°) and C (with a mean slope of 32°) unstable slope angles were lower than that of the stable slope (40-60°), but Type A slope angles (mean slope is 54°) are similar to those of the stable slope. This result

was verified in field investigation (please see figure 4a). We added this part in L254-258.

[Figure]

Figure 3

This result is contrary to the general conclusion of the landslide study-the steeper the slope, the stronger the shear stress, and the lower the Factor of safety (McColl, 2015). Why is that so? After carefully studying the similarities, differences, and correlations of three types of unstable slopes, we proposed a conjectural paraglacial slope mass transport model of HLG (figure 4b).

After the glacier was downwasting from the initial state (stage I), the upper of the steep moraine slope quickly destabilized, making the entire slope slow, corresponding to funding in van Woerkom et al. (2019), Ballantyne (2013), and Cully et al. (2006); then the moraine slope slowly slides down (stage II). During the exposure of the moraine, vegetation may be colonized, or gullies may be formed by debris (or water) flows washing away (stage II a); when the gullies gradually expand, headward erosion may occur in the upper of the gullies (stage II b). Sediments at the base of the slope or fell onto the glacier surface are transferred again as the glacier moves (stage III); until the slope is steepest when moraine has been removed and bedrock is completely exposed to the ground and may collapse under other disturbances (stage IV).

We have added this part in L566-575.

[Figure]

Figure 4

[20] L395-402: The role of vegetation colonisation is (reasonably) discounted for stabilising the Type B failures that involve deeper-seated sliding, but what about the role of vegetation colonisation

for stabilising other types of erosion process in the valley? Do you see a reduction in say Type C hillslope responses over time? Again, this might make for another comparison/contrast with the Eichel et al (2018) model.

Sorry, we didn't see that kind of reduction in Type C, at present. And we removed the part of a comparison with the Eichel et al (2018) model.

[21] L400-403: It is a shame that there was not more effort made to understand why all Type B sites appeared to increase in movement rate between 2017-2018. What further analysis could be done to explore this? Did Type C response (i.e. gullying) also increase? Can the data from Figure 3 be used to analyse this further? Did any slopes downstream of the glacier terminus show any increases in movement or erosion during this time (i.e. helping to rule out glacier thinning as a cause)? Did the upper parts of the slope failures speed up to the same extent as the lower parts (perhaps more suggestive of rainfall as a driver) or did the lower part speed up the most (perhaps suggesting removal of toe support from ice thinning)?

Thank you for your comment! The abundant precipitation and low temperature in 2018 might be the main reason for the occurrence of 2018 rockfall and the increase in movement rate of Type B. And unfortunately, due to the steep terrain of HLG, the UAV imagery were not covered the whole PSFs area, meanwhile, due to the limitation of the resolution, the RS images from 2016 to 2019 do not change significantly in Type C region, so it is hard to calculate its erosion rate in that period. In addition, the time period of figure 3ab is 1966-2016.

Please see our new discussion part *5.2 Paraglacial geomorphology process in deglaciating monsoonal temperate environments.*
* * *
L19: the term 'slope slide and collapse' is inconsistent with standard mass movement terminology. Consider revising (e.g. see Hungr et al., 2014 classification)

Corrected.

L37: remove 'and' after '(4)' since this is not the end of the list.

Corrected.

L96: the statement 'stretching to the Little Ice Age (LIA) end-moraine' seems to contradict the later statement on L100 that the 'HLG has retreated more than 2 km since the LIA'. Please reconcile.

Corrected.

L104: do you mean 'proglacial' instead of 'preglacial'?

Yes, we corrected it.

L113-117: perhaps this hazards rationale could be moved to or restated in the introduction.

It's okay to put it here and sublimate it in this section.

L129-130: consider improving clarity of statement 'NDVI can eliminate a part of the effect of hill-shade…' (i.e. explain more clearly).

Corrected.

L132: consider different choice of words: 'less suffered from' L134: what is meant here by 'partly'? L137: add 'and' before '19 August…'

Corrected.

L196-198: consider revising sentence 'Field observations indicate…' This is unclear and needs further elaboration.

Corrected.

L240: you describe the slope failure as coming from a 'south-west facing slope' but on L244 refer to a 'steep north-facing slope'. Are these the same failure, and if so, is there a mistake with the latter?

Corrected.

L247-249: it is hardly a surprise that rockfalls are able to reach the glacier and become supraglacial debris. Perhaps this sentence adds little and can be removed?

Corrected.

L259: this sentence 'each with a mean area of 750 m2' reads that all 15 rockfalls had the same area. But I presume 750 m2 was the mean area, and that they each had different sizes? Perhaps provide the size range.

Corrected.

L283: what type of 'analysis'?

Corrected.

L329: what is 'an increase in sediment-covered area' referring to? Is this the supraglacial sediment cover?

Corrected.

L409-410: what is meant by 'sediments were transported along the transport'? What is 'with a fast erosion' referring to? 'largest change in area of C1 travels about 150 m…' is unclear. Confusing use the term 'travels'.

Corrected.

L415: consider revising text 'seasonal plus of enhanced or uprush erosion rates'. This is unclear.

Corrected.

L448: 'a large number of debris fell into the glacier surface' From which process? Rockfall, debris sliding, etc?

Corrected.

L486: this is the first mention of 'debris avalanching' so it should not appear in the conclusion for the first time.

Corrected.

**Response to 'Comment on esurf-2021-18' from Jan Henrik Blöthe**

We thank Dr. Blöthe for his very constructive and helpful comments which will clearly help to improve our manuscript by pointing out weaknesses but also suggesting the respective improvements which we will be necessary to be included for the next version of the manuscript. We try our best to provide as many details as possible to better reply to the reviewer. Please find all the details below.

[1] Overall, the manuscript is well written, but sadly lacks rigorous identification of previous work and the work conducted here. Especially in chapter 3 (Methods and data), the authors need to put more effort into making it very clear, which information has been obtained from earlier studies and how the data produced in the framework of this manuscript was produced.

We thank you for pointing out the disadvantage of our manuscript and we have made it clear in the new manuscript.

[2] A very basic and important point that the authors fail to include in their work is a thorough error assessment of all measurements they conducted. This is neither addressed in the methods section, where it remains rather unclear how exactly mapping has been conducted, nor are the errors associated with mapping (based on visual interpretation?) in remotely sensed imagery discussed later in the text. For

Sorry, we did not calculate the error on the visual interpretation and bare ground automatic extraction. Such errors are usually estimated by ±0.5 pixel uncertainty along with the boundary shape, we have added the error estimation in the new manuscript L154-157.

[3] Chapter 3.2: Here the authors mainly present details of earlier studies that quantified glacier mass balances changes of the HLG. I would recommend to include these background information into the description of the study area (Chapter 2) and shift the focus of Chapter 3.2 to the method applied here.

Thank you for your advice, we have moved it.

[4] From the technical description in L158-164, I take that the authors used three digital elevation models (DEMs), namely "the TopoDEM (1966), the Shuttle Radar Topography Mission (SRTM) 30m DEM (2000) and the ASTER-DEM difference between 2000 and 2016". While I am familiar with the latter two (the ASTER DEM data is from Brun et al. 2017, include reference here), I am not aware of the TopoDEM (1966). If this refers to the DEMs calculated in Cao et al. (2019), this needs to be indicated here. Moreover, from the technical details outlined here, it remains unclear to me whether the authors only analyzed five profile lines, or calculated full DEMs of difference and used five profile lines to visualize the data. Please elaborate this in more detail and outline, in the case data was only analyzed along five profile lines, how these are encompassing the full variability of surface changes on the glacier tongue.

TopoDEM (1966) is not calculated by Cao et al. (2019), but has been reported by Liu et al. (2010) and Zhang et al. (2010). We calculated full DEMs and used five profile lines to visualize the data. We added more details in L179-181 to make this information clearer.

These profile lines were selected to try to cover every type of slope and keep a certain distance

between the transverse profiles so that the extracted results are better for comparison. We added this information in L184-185.

[5] Furthermore, in L165-174 (still Chapter 3.2), the results of earlier studies that quantified ice flow dynamics are presented. In the final sentence the authors describe a comparison of "long-term ice flow velocity changes […], based on three-periods results of 1982-1983 (in situ observed), 2007-2011 and 2014-2018 (SAR satellite derived)." Where does this data come from? Is this an analysis done by the authors, or does it refer to the studies presented before? As the dates mentioned here do not match the time spans given for the studies cited above, I would ask the authors to either clearly indicate the provenance of these data.

The *"1982-1983 (in situ observed)"* data are from L165-168 '*the earliest observations of surface velocity of the HLG were during 1982-1991*'. Here we used the data from the earliest period (1982-1983).

The *2007-2011 and 2014-2018 (SAR satellite derived)* data are from L170-171 '*acquired by PALSAR-1/2 satellites from 2007 to 2018, Liu et al. (2019a)*'. Here we used the data from the two periods (2007-2011 and 2014-2018).

We have marked the source of the data in parentheses, and we added a scanned picture of the 1982-1983 published velocity map in the supplementary files (Fig. S4).

[6] Chapter 3.3: This is a very brief description of how slope movement was quantified, given that large parts of the results and discussion build upon this data. While manual tracking of tie points in repeated imagery can be considered a fairly robust technique, I would recommend that the authors at least try to quantify the error associated with this tracking of tie points. This can be easily achieved by tracking stable surfaces in the vicinity of the slope failures. Furthermore, may I suggest to include the individual vectors for manually tracked tie points in Fig. S2? Last but not least, let me point out that there are multiple software solutions that allow for automated tracking in consecutive imagery, which would result in full 2D velocity fields that might enable a much deeper insight into the mechanisms of the failures (and reveal local variability).

This is a good suggestion. We included the individual vectors for manually tracked tie points in Fig. S3.

Indeed, there are multiple software that can automatically track in consecutive imagery, and we have tried it. However, because the vegetation on the Type B slope is well-established and fewer tie points can be identified, the automatic tracking effect is not good, we therefore chose manual tracking with higher accuracy.

[7] Section 5.1.1: Here the authors discuss the possible preparatory and triggering factors for the rock fall they observe during their study period. The authors might want to further elaborate how the exceedance of a precipitation threshold of 60 mm for a single day in June 2018 is connected to the triggering of a rock fall in October 2018. This is mainly referring to L362-65 and Tab. 3, where the authors argue for a precipitation intensity anomaly, which I find hard to follow given the data. Yes, 2018 has seen one day with more than 60 mm of daily rainfall, i.e. 61 mm (L364). Without giving more details on the exact precipitation values for 2016 that also saw three days with more than 40 mm per day, I do not think this can be seen as an indication for a precipitation intensity

anomaly. In my view, it would be worth to look at the antecedent rainfall in the five to ten days before the failure and compare antecedent rainfall statistics between years, especially in a setting with very few days without rainfall. Furthermore, in section 5.1.1 the authors argue for a potential triggering by frost action. It is my feeling that also this remains speculative, as the cold interval the authors refer to here (01-09 October 2018; L369, Fig. S4) happened at least 5 days before the failure that the authors date to 15 October 2018. However, the temperature data for 2018 is unavailable for the days following 09 October, making this link questionable.

Thank you for your comment! We can understand your opinion that the precipitation in 2018 is not considered abnormal. We will revise the description of this part in the new version of the manuscript. And we revised the study time to five to ten days before the rockfall occurred (**5-20 Oct 2018**) and added the below figure in the manuscript.

[Figure]

The mean daily precipitation between **5 and 20 Oct 2018** is 5.89 mm which is significantly higher than 2014, 2015, and 2017, although not significant compared with 2016, the number is still high.

As for the temperature data after 9th October 2018, we obtained and checked the daily temperature data of four periods (2 am, 8 am, 2 pm, 8 pm) by manually observation at the 3000m observation station at Mt. Gongga (http://ggf.cern.ac.cn/meta/metaData), and calculated and plotted the daily mean temperature curve, which has been added to Fig. S4 (the purple dash line in the below figure). We revised it in the manuscript.

[Figure]

Note: MO means manual observation.

[8] L35-40: Here the authors gather five modes of response to slope failure, though I have the feeling that mode 5 "paraglacial debris cones and valley fills" is rather the results, i.e. deposit of the processes listed in 1-4.

We thank you for pointing out this error and we have corrected it.

[9] L67-68: Not really relevant at this point, as the tourism activity at the site is detailed in L109-117.

We removed it.

[10] L101-02: In L152-53, this information is from Zhang et al. 2010, please add reference here

Thank you, we have added it.

[11] L124: How was the glacial area mapped exactly? As the HLG is debris-covered, a precise delineation between debris-covered ice and the debris-covered surroundings is not trivial. Please elaborate in detail, how mapping of glacier extent was conducted and what the associated uncertainty of this mapping was.

Due to the complex terrain of HLG ice surface-numerous ice cliffs and crevasses- as well as the high slope on both sides of the glacier, it is very difficult to use GPS to measure the glacier boundary on the ground. Therefore, at present, the use of the medium- and high-resolution remote sensing images has become the most effective method of glacier boundary extraction. Although the boundary of the debris covered glacier is not as distinct as the clean glacier, it is still visible. As mentioned above comment [2] the errors are usually estimated by ±0.5 pixels.

The following figure takes PL image as an example:

[Figure]

[Figure]

[12] L124-26: Also for the paraglacial slope failures (PSFs), it is not clear how exactly these were mapped. In L121-24 it is described that using the NDVI, vegetation covered areas were excluded. But how exactly was the mapping of the PSFs achieved in remote sensing imagery. What were the criteria for mapping? Was this mapping field-evidence based and if so, when was field-work conducted?

[13] L130-34: Again, it remains unclear at this point how PSF boundaries were extracted, what validation means in this respect and based on which criteria manual correction was done.

As mentioned before, the current method does have the disadvantages you indicate, but it can delineate the expansion of exposed slopes in most areas. We discussed the uncertainty of the delineation boundary in L154-157. Manual correction of PSF is mainly based on the visual interpretation of the RS image, eliminating the pattern and obvious mis-extraction (relying on empirical judgment).

[14] L138: I would suggest to state this in more detail. At this point, it is unclear, whether the "mean quality of 0.01 m" refers to the position accuracy of the RTK UAV, or to the SFM output. Furthermore, "+1 ppm (RMS) in XY" is neither clear in this regard. In order to allow the reader to follow and to judge the quality of the data used in this study, I suggest to explain in detail, how the UAV images were processed and what the residual mismatch of their geolocation is.

The GNSS quality information is required from DJI official website (https://www.dji.com/uk/phantom-4-rtk/info#specs)

*Horizontal 1 cm + 1 ppm(RMS)*
*1 ppm means the error has a 1mm increase for every 1 km of movement from the aircraft* (we added this sentence in L139-142)

UAV measurement is a very mature technology, albeit a new one; we described our UAV mappings in L139-154 with some more details.

[15] L142-43: I take it that the authors co-registered and orthorectified the UAV-derived orthomosaics with the PALSAR DEM, or is the sentence correct that individual UAV images were used for this? May I ask the authors to elaborate this a bit more, as this is cionfucing? IN line XX you write that these are already orthorectified?

We use PL images and ALOS PALSAR DEM to co-register and orthorectify the UAV images which are synthesized by ContextCapture Center Master Software based on UAV photographs. Due to the

steep terrain of HLG, DEM correction is required for the synthesized images. We added more detail in L147-150, thank you.

[16] L156: The authors compiled a data set here, but I fail to see where this data set is included in the manuscript? Is there a figure or table that shows this compilation?

We used the dataset in Figure 3 for ice thinning (elevation change) analysis.

[17] L174: In the very brief section following this heading, the "outline change rate" is not mentioned nor explained how this is quantified. Instead, the authors quantify the rate of headscarp erosion. Consider rephrasing the heading here.

Ok, thanks for mention it, we changed the heading to "Slope movement and headscarp erosion rate".

[18] L178-180: May I suggest to elaborate more clearly how the outlines of failures were used to calculate a mean annual retreat rate for headscarps?

We calculated the mean distance between the two outlines where the headscarp erosion occurred at the head of the slope. It can be obtained by using the Average Nearest Neighbor tool in QGIS. We added this sentence in L171-172.

[19] L220-23: The time spans given here in the text are not the same as in Fig. 4. What is the rate between 2000 and 2019?

What I want to express here is that the growth rate of EBGA (now we change the term to PBGA) has increased from ~0.01 km$^2$ a$^{-1}$ at the beginning (1990-2000) to 0.1 km$^2$ a$^{-1}$ now (2019-2020).

[20] L224: Which area are you referring to here?

Sorry to confuse you, but what I want to say here is the area of EBGA (now we change the term to PBGA).

[21] L225-26: Where is the data for the lower frequency of PSFs?

Sorry, we have revised it.

[22] L227: Where does the knowledge of slope material come from? Did you map out slope material distribution during field work?

We revised it, please see section *3.1 PSFs mapping and classification*.

[23] L239-44: This description of the rock fall (PSF type A) needs to be refined. In L240 it is stated that the rock fall occurred on a south-west facing slope, while in L244 it is stated that the mass detached from a steep north-facing slope? Is it that the general topography is south-facing and the nice of the detachment faces north? The picture in Fig. 5b, however, does look like the source is also facing the glacier – please clarify.

Sorry, we have revised it in L260-262.

[24] L241-42: This is not precise enough. How can a deposit of a rock fall be "450m in height from the glacier surface" and at the same time, "cover a height of 380 m"?

We revised it, please see L260-262 & 264-266.

[25] L258-59: Please indicate what magnitude is considered small here, as to me "each with a mean area of 750 m2" is not clear.

We revised it, please see L274-275.

[26] L262-63: Please try to be consistent in labelling the processes: "Sediment-mantled slopes slide and collapse" vs. "Sediment-mantled slope slide and collapse". In my view, both seem a bit clumsy – you might want to rephrase.

Thank you for pointing out that, we changed it to "Sediment-mantled slope slide" based on "Colin K. Ballantyne, Paraglacial geomorphology, Quaternary Science Reviews, Volume 21, Issues 18–19, 2002, https://doi.org/10.1016/S0277-3791(02)00005-7, and Hungr, Oldrich, et al. "The Varnes Classification of Landslide Types, an Update." Landslides, vol. 11, no. 2, 2014, pp. 167–194."

[27] L263-66: The numbers given here are surprisingly round and do not match the sums of the individual numbers mentioned in Tab. 2. Also, in Tab. 2, there are no errors associated with the numbers given for the area. Please include a statement on the precision of these estimates in the text. This also applies to L310.

The four Type B slope areas in Table 2 add up to 371673m², which is approximately 370,000. We added "~" in L282

The number "297,000 m²" is the total increase area from 1990 to 2020 calculated through satellite images, which is not the one calculated from UAV imagery (2016-2019). We added some details in L285. Thank you for pointing it out.

[28] L271-75: This comes as a surprise here and rather belongs to the discussion.

Thanks for mention, we have corrected it.

[29] L276-78: How was the error of 0.04 cm d-1 quantified?

From L154 "a *mean horizontal error of 0.15 m*", error automatically calculated by ground control points during the UAV registration procedure.

[30] L283: How do the authors know that the detectable slope movement began around 2000? Has this been published, or did the authors run additional analysis beyond the 2016-2019 UAV surveys that have been described in the text. Same applies for L296.

According to the historical remote sensing images (2000 TM, 2002 Google earth, 2011, 2013 RapidEye, please see the below figure), the deformation of the slope began to appear in 2000.

[Figure]

[31] L286: "the landslide has fallen" sounds as if it was a vertical movement that was quantified? Until now, it is my understanding that the authors quantified 2D horizontal displacement.

Yes, the landslide was quantified only in 2D horizontal directions. The sliding rate could be estimated based on the surface slope of the terrain. Thanks for mention, we have corrected the wording in L301.

[32] L400-02: This comes as a surprise as a) in L361-62 the authors argue that 2018 has seen relatively low temperature. Furthermore, Fig. 9 suggests that 2017 has a data gap in temperature readings.

Thanks for mention, we corrected it.

We used manual observations to supplement the mean temperature data for 2017 (L202-204).

[33] L430-38: I am not sure how this is connected to the data and topic presented in this study? Consider removing paragraph or elaborate the connection to paraglacial slope adjustment.

Thank you for your suggestion! We added more detail to make the link between glacier velocity and hillslope processes clearer.

The slowing of the glacier velocity and the glacier thinning corroborate each other. On the one hand, the thinning of the glacier slows down the glacier velocity; on the other hand, when the glacier velocity slows down, the ice flux transported from upstream to downstream is reduced, which accelerates the glacier thinning and ultimately leads to the slope sliding.

[34] L444-46: Is this supposed to be a general statement? Also, check grammar.

Thanks for mention, we corrected it.

[35] L448-50: It is hard to see how the deposition of debris onto the glacier is directly affecting climate. As the authors try to outline in Figure 8, debris cover generated by slope failure might have an effect on glacier downwasting, which in turn can have a tiny effect on the climate.

Yes, as shown in Figure 7, debris cover generated by slope failure might have an indirect effect on the climate.

L526: We will change the sentence to "…which in turn affected the surface energy balance for melting and therefore downwasting rate of the glacier, which will thus influence the rate of runoff generation and its contribution to the sea level rise from glacierised catchments."

[36] L464-70: While McColl and Davies (2013) showed that also failures of similar magnitude as B2 in the present study can deform glacier-ice at the rate of mm/yr (assuming a min. average thickness of ~1 m), the evidence presented here is limited and the discussion of this aspect remains too surficial. It might be a way forward, to present the displacement data in more detail and quantify the supposed narrowing and squeezing effect that is mentioned here to back this aspect with data.

As reply in comment [6] above, we included the individual vectors for manually tracked tie points in Fig.S3 according to your suggestion.

[37] L474-75: What data is this statement based on? Is there a study that found this increase in debris-flows and flash-floods that could be cited here?

Yes, there is a study that can be cited here: "Lu, R., and Gao, S. (1992). Debris flow in the ice tongue area of Hailuogou Glacier on the Eastern slope of Mt. Gongga. Journal of Glaciology and Geocryology, 73-80". However, there is no data and literature to support the increase in the frequency of debris-flows and flashfloods, we removed "*increased frequency of*", and cite the literature in the text, thanks for mention.

[38] L482: In L195-96 the authors state that between 2016-2019 the glacier terminus retreated by >150 m with a rate of ~52 m yr-1. Which is true?

I'm sorry, it's a mistake. We have checked all the numbers in the new version.

[40] L485-86: In your data, type A landslides are limited to one rock fall. Did you find evidence for debris avalanches as well?

No, we didn't find any debris avalanches.

[41] Figure 1: Might I suggest to add the glacier outline for the years 1982/83, as in the text the authors have quantified (or cited) the displacement values for this period (Fig. 3).

Ok, we have added it.

[42] Figure 3: Is there a reason for the black line being thinner in D-D` of the first column? Also, in the caption, replace annual thinning rate with average annual thinning rate, as these have been quantified over multiple years, right? What remains unclear is the provenance of the data shown in the right column. Has this data been calculated from remote sensing data by the authors? If not, add the references to the data to the caption.

Thank you for your careful review. We have corrected the D-D 'line as shown in the following picture and corrected the caption and added the reference.

[43] Figure 4: Change "(white line)" to "(blue line)" in the caption, as this is showing the glacier outline, judging from the legend.

Thanks for mention, we have corrected it.

[44] Figure 6: Might I suggest to label the vertical axis of the last column with "displacement velocity"? Slide velocity implies a vertical component, but these are horizontal displacement values, right? Furthermore, in the caption, what does the reference to Qiao Liu et al. imply? Has this data been published before? I cannot find this reference in the reference list.

Thanks for mention, we corrected the vertical axis to "displacement velocity". This data has not been published, it was requested by the ESD journal to include the source in the maps and photos. The UAV images were taken by Qiao (Corresponding author of this manuscript) et al and therefore need to be noted here.

[45] Figure 8: May I suggest to give this figure a more detailed caption that explains the arrows and the reciprocal effects? Also, why does Type C have a larger arrow than Type A and B, what exactly does "remarkable glacial debuttressing" imply? Also check grammar of text box PSF Type A.

We added more detail to the caption. The three arrows of type A-C on the right end are just for the sake of aesthetics, nothing else, and we have revised it.

[46] Figure 9: Here it should be indicated over which time window the moving average for temperature and precipitation has been calculated. Furthermore, the reason for and length of the temperature data gap in 2017 should be mentioned in the caption. Also, are Tem_smooth and Pre_smooth appropriate abbreviations for a running mean of these variables?

Thanks for mention, we have corrected it.
* * *
L130: delete "can"

Corrected.

L132: "generally suffered less from" (?) instead of "are generally less suffered from"?

Corrected.

L136-140: very long and in my view incomplete sentence; check grammar

Corrected.

L166-67: check grammar

Corrected.

L196-98: please rephrase

Corrected.

L228-29: broadly coincide with?

Corrected.

L239-41: "rock fall event […] occurred around October 15, 2018"? Check grammar

Corrected.

L246-249: check grammar

Corrected.

L250: "can thus can be some clearly"?

Corrected.

L256: delete "are"

Corrected.

L259-61: please rephrase, not clear what exactly this refers to and how relevant a mean distance to the glacier is in this context.

Corrected.

L301-04: check grammar

Corrected.

L309: seasonal snowmelt

Corrected.

L 310: Numbers in Tab.2

Corrected.

L335: glaciofluvial sediment instead of glaciofluvially transported glacial sediment

Corrected.

L362: You should maintain the figure order! Figure 8 has not been referenced in the text and is only done so in L450.

Corrected.

L361-62: There is no temperature given for 2017 and both 2014 and 2015 seem to be below 5°C, so I don't think you can make this statement

Corrected, we added the 2017 temperature.

L373: freezing of water begins?

Corrected.

L389: delete hosting

Corrected.

L409-10: check grammar

Corrected.

L414: check grammar

Corrected.

L415: uprush? Can this be used in this context?

Corrected.

L418: HLG

Corrected.

L441: obvious compared to?

Corrected.

L448: replace into with onto

Corrected.

---

## Author Response (AR2)

Dear Editor,

Thank you for taking the time to review our manuscript. We have carefully revised the abstract, introduction, discussion, heading, and formal requirements. We hope that the revised manuscript is more comprehensible. Please see our highlighted part of the revised manuscript.

With kind regards,
On behalf of all authors
Yan Zhong, Qiao Liu, and Matthew Westoby